# Evaluating Greenland Surface-Mass-Balance and Firn-Densification Data Using ICESat-2 Altimetry

Benjamin E. Smith[1], Brooke Medley[2], Xavier Fettweis[3], Tyler Sutterley[1], Patrick Alexander[4,5], David Porter[4], Marco Tedesco[4,5]

[1]University of Washington Applied Physics Laboratory Polar Science Center, Seattle, WA, 98122, USA
[2]Cryospheric Sciences Laboratory, NASA Goddard Space Flight Center, Greenbelt, MD, 20771, USA
[3]SPHERES research unit, Geography, University of Liège, Liège, Belgium
[4]Lamont-Doherty Earth Observatory, Columbia University, Palisades, NY 10964, USA
[5]NASA Goddard Institute for Space Studies, New York, NY 10025, USA

Correspondence to: Benjamin Smith (besmith@uw.edu)

**Abstract.** Surface-mass-balance (SMB) and firn-densification (FD) models are widely used in altimetry studies as a tool to separate atmospheric-driven from ice-dynamics-driven ice-sheet mass changes, and to partition observed volume changes into ice-mass changes and firn-air-content changes. Until now, SMB models have been principally validated based on comparison with ice core and weather-station data, or comparison with widely separated flight radar-survey flight lines.

Firn-densification models have been primarily validated based on their ability to match net densification over decades, as recorded in firn cores, and the short-term time-dependent component of densification has rarely been evaluated at all. The advent of systematic ice-sheet-wide repeated ice-surface-height measurements from ICESat-2 (the Ice Cloud, and land Elevation Satellite, 2) allows us to measure the net surface-height change of the Greenland ice sheet at quarterly resolution, and compare the measured surface height differences directly with those predicted by three FD/SMB models: MARv3.5.11

(Modèle Atmosphérique Régional version 3.5.11), GSFCv1.1, and GSFCv1.2 (the Goddard Spaceflight Center FD/SMB models version 1.1 and 1.2) . By segregating the data by season and elevation, and based on the timing and magnitude of modelled processes in areas where we expect minimal ice-dynamic-driven height changes, we investigate the models' accuracy in predicting atmospherically-driven height changes. We find that while all three models do well in predicting the large seasonal changes in the low-elevation parts of the ice sheet where melt rates are highest, two of the models

(MARv3.5.11 and GSFCv1.1) systematically overpredict, by around a factor of two, the magnitude of height changes in the high-elevation parts of the ice sheet, particularly those associated with melt events. This overprediction seems to be associated with the melt sensitivity of the models in the high-elevation part of the ice sheet. The third model, GSFCv1.2, which has an updated high-elevation melt parameterization, avoids this overprediction.

## 1 Introduction


Ice-sheet surface heights vary on time scales from hours [*Amory et al.*, 2021; *Lai et al.*, 2021] to millennia [*Dahl-Jensen et al.*, 2013; *Khan et al.*, 2016]. Repeated altimetry measurements can provide estimates of ice-sheet mass changes [*Shepherd et al.*, 2020], and thus their contribution to sea-level change, providing clues to the mechanisms driving mass loss [*Catania et al.*, 2020; *Smith et al.*, 2020] based on spatial patterns and timing of the changes. On an ice sheet in steady state, whose volume
and mass are constant in time, snow accumulation and ice ablation at the surface are balanced by ice-flux divergence in the ice-snow column (e.g. thinning of the ice column related to horizontal stretching of the ice), and by snow and firn compaction in the near-surface layers. Any deviation of the rate of one of these processes from its steady-state rate will result in a non-zero rate of surface height change. We expect to see large variations in the net surface mass balance over the course of a year, and, over most of the ice sheet, to see much slower, annual-to-decadal variations in the rate of ice-flux divergence driven by
evolution of the local stress balance of the ice. Thus, even in a part of the ice sheet where the climatological mean surface mass balance exactly compensates for ice flow, we expect to see seasonal surface-height variations. Secular trends in the local net ice sheet mass balance, such as thickening due to net annual SMB that exceeds the local flux divergence, or thinning due to increased ice-flow speeds that are not balanced by additional snowfall, are superimposed on these seasonal signals.

Time series of ice surface height measured by altimeters cannot, by themselves, distinguish between the effects of
surface-mass-balance changes and those of variations in ice flow, or between surface height variations caused by changes in the average firn density (e.g. due to imbalances between snowfall and compaction) and those caused by changes in the ice-column mass. This leads to two sets of challenges in the interpretation of altimetry records from the ice sheet: The first is in understanding the relationship between ice volume changes and ice mass changes, which is complicated by variations in near-surface density. The second is in understanding whether ice mass changes are driven by changes in ice dynamics, such as
thinning driven by the acceleration of outlet glaciers, or by variability in surface mass balance. These challenges may be addressed in part using surface-mass-balance (SMB) and firn-density (FD) models. SMB models provide estimates of the variability in accumulation, melt, and runoff, which allow estimates of the contribution of atmospheric processes to ice-sheet mass change. FD models are driven by information about heat and moisture flux variability provided by SMB models, and provide estimates of variability in the firn-air content (FAC) as a function of time and depth; the difference between the total
measured volume change and the total FAC change gives the change in the ice mass, which can be converted directly into ice mass change. In some of the most rapidly changing parts of the Greenland ice sheet (i.e. outlet glaciers and the regions immediately upstream), height variations are driven in large part by changes in ice velocity (thus flux divergence rate changes) [*Moon et al.*, 2015]. These areas, however, are limited to a zone near the coast extending a few tens of kilometers inland; over the majority of the ice sheets, ice velocity has been relatively constant since the first systematic measurements in the late 1990s.
In the absence of large variations in velocity, most ice-elevation changes should be SMB and FD driven and all height change components can, in principle, be described by a combination of FD and SMB anomalies.

A variety of models are available that can generate SMB (e.g. [*Fettweis et al.*, 2017; *Gelaro et al.*, 2017; *Noel et al.*, 2015]) and FD (e.g. [*Brun et al.*, 1989; *Kuipers Munneke et al.*, 2015; *Stevens et al.*, 2020]) estimates for Greenland and Antarctica, each with differing temporal and spatial resolutions, different internal representations of the physical processes driving SMB and firn densification, and driven by different climate-forcing data. Some processes within SMB models (e.g. surface albedo evolution) can be tested by comparison with remote sensing data (e.g. [*Banwell et al.*, 2012]), and SMB models have been tested by comparison with point measurements, such as automatic weather stations, ice cores, and ablation stakes (e.g. [*Noel et al.*, 2015]), by comparison with accumulation estimates derived from layering observed in ground-penetrating radar data (e.g. [*Koenig et al.*, 2016; *Medley et al.*, 2014]), and, in bare-ice zones, by direct comparison with altimetry data [*Sutterley et al.*, 2018]. Densification in FD models has been tested and, in some cases, calibrated by comparison with ice core density profiles (e.g. [*Alexander et al.*, 2019; *Li and Zwally*, 2015; *Ligtenberg et al.*, 2011; *Lundin et al.*, 2017; *Munneke et al.*, 2015]), borehole measurements [*Hawley et al.*, 2020; *Morris and Wingham*, 2014], and, for a limited set of measurements in Antarctica, by repeated radar measurements [*Ligtenberg et al.*, 2015]. We have identified one study [*Verjans et al.*, 2021] that has used altimetry differences to validate combined SMB and FD models in Antarctica, and a second [*Munneke et al.*, 2015] that used altimetry differences to evaluate trends in snow-surface heights predicted by models in Greenland.

Previous model evaluations, particularly those of the FD models, have been limited in their spatial extent, and do not demonstrate how the accuracy of the models vary over the full range of ice-sheet surface conditions and seasons. In this paper, we present an evaluation of three SMB/FD models in Greenland based on height changes measured with NASA's ICESat-2 satellite between the autumn of 2018 and the end 2020, a period that includes the substantially anomalous summer-2019 melt season [*Tedesco and Fettweis*, 2020]. Although combined SMB and FD models can be evaluated at regional scale in studies that evaluate ice-sheet mass balance based on multiple redundant datasets (e.g. [*Martin-Espanol et al.*, 2016]) including gravimetry and altimetry, the coarse spatial resolution of these studies means that the effects of velocity changes are not as easily separated from SMB-driven changes in these data as they are in the altimetry measurements. The high (cm-level) vertical precision, 100-meter spatial resolution and quarter-annual temporal resolution of the ICESat-2 data allow us to make pointwise comparisons between the behaviour predicted by the models and the measured height differences, and, by selectively isolating groups of difference data in which the models predict different SMB processes to play a strong role in surface height change, we evaluate the accuracy with which the models can predict these processes. Our results offer an ice-sheet-wide view of the accuracy of model processes driving surface height changes.

## 2. Data and methods

Our results are based on altimetry data from ICESat-2, selected based on ice-surface velocity data. We compare these data with height-change predictions based on modelled SMB and FD changes from two atmospheric models, driving three different FD models. We describe each below.

## 2.1 Altimetry data

Our altimetry data are derived from the ATLAS (Advanced Topographic Laser Altimeter System) instrument on board NASA's ICESat-2 satellite. ATLAS measures the height of the ice sheet surface using six laser beams, which measure three pairs of tracks, each separated from its neighbour(s) by 3.3 km. The central pair follows a set of 1387 reference ground tracks (RGTs), which are separated by about 10 km in central Greenland (70°N), and the 3.3-km offsets between the reference pair tracks (RPTs) followed by the right, central, and left beam pairs help fill in the gaps between RGTs. This orbital and beam geometry gives pair-to-pair sampling better than 3.3 km for most of Greenland and Antarctica.

The beams within each of ICESat-2's beam pairs are separated by ~90 m, allowing each pair to uniquely determine the cross-track surface slope. The satellite makes repeat measurements on each RGT, once every 91-day cycle. ICESat-2 began making measurements in October, 2018. For the first two 91-day cycles, the on-board software to point the central beam pair at the RGT was not configured correctly, so ICESat-2 measured tracks displaced from the RGTs by up to several km. This problem was corrected at the start of cycle 3, in April, 2019, and subsequent cycles of data followed the RPTs with a precision better than 10 m [*Luthcke et al.*, 2021]. The ICESat-2 dataset continued uninterrupted to the present (September, 2021) except for a 14-day period in between June 26 and July 19, 2019, when the instrument was shut down because of problems with the satellite orientation.

Elevation-change data in this paper are based on release 004 of the ICESat2 ATL11 data product [*Smith et al.*, 2021] which combines measurements from multiple cycles to correct for the spatial variation in surface height around each RPT. The limited precision of ICESat-2's repeat track pointing introduces small apparent height differences between measurements from different cycles, with a magnitude approximately equal to the product of the across-track offset and the surface slope. At each of a set of reference points spaced every 60 m along the RPTs, the ATL11 algorithm solves for a reference surface that corrects for these offsets to give height estimates for each cycle with little or no contribution from the across-track offset. It uses the same correction for points where tracks from different cycles cross the RPTs (crossover points). For this study, height differences since the beginning of RPT pointing (April, 2019) are calculated based on height measurements along the same RPT. Height differences from cycles 1 and 2 (prior to April, 2019) are calculated based on crossover-difference measurements between the early non-RGT-pointed measurements and the cycle-3 (and later) RGT-pointed measurements.

ATL11 provides two kinds of error estimates. Per-point estimates (*h_corr_sigma*) include the errors related to the accuracy of the reference surface and the precision of the ICESat-2 range estimates, that are uncorrelated between adjacent reference points. Systematic error estimates (*h_corr_sigma_systematic*) include the contribution of uncertainties in measurement geolocation and the satellite's radial orbit errors to the measurement errors. In the interior of the ice sheet, per-point errors tend to be on the order of 1-2 cm, becoming somewhat larger for coastal areas where rougher surfaces and larger slopes degrade the precision of the instrument. Correlated errors are roughly proportional to the measurement geolocation uncertainties times the surface slope. However, because release-004 along-track products use nominal, pessimistic estimates of the geolocation errors (20 m in each direction), and studies that assessed the accuracy of release-003 products found that

they in fact had smaller geolocation errors, generally less than 6.5 m [*Magruder et al.*, 2020], we expect to see correlated errors ranging from a few cm in the interior to ~0.65 m in the most strongly (~10%) sloping areas near the coasts. We find that typical RMS error magnitudes for the subsamples of data we present here are on the order of 3 cm for the inland part of the ice sheet, and on the order of 10 cm for coastal regions.

We present the ICESat-2 data as eight epochs of height differences, all except the first made up of differences between subsequent 91-day cycles. Because cycles 1 and 2 were not collected on the RGTs, the first two epochs use crossover differences relative to cycle 3, thus the first epoch is made up of differences between the fourth quarter of 2018 (18.Q4) and the second quarter of 2019 (19.Q2), and subsequent epochs are made up of differences between adjacent quarters (e.g. the second epoch is 19.Q1 to 19.Q2).

**2.1.1 Velocity-variability-based data selection**

This study is intended to evaluate the accuracy of the representation of SMB-driven processes on the Greenland ice sheet. In parts of the ice sheet where the ice-flux divergence is out of balance with SMB, we expect to see surface-height changes due to a combination of ice-dynamic changes and SMB changes. Because we cannot accurately predict the magnitude of height variations associated with surface velocity variations, we restrict our analysis to areas of the ice sheet for which the

temporal ice-flow variability is small (less than 20 m yr$^{-1}$ deviation from the multi-decadal mean, with seasonal variations less than 10 m yr$^{-1}$, see SOM, Fig. S2). This approach removes much of the lowest-elevation portion of the ice sheet, and part of the upper catchments of a few of the fastest-changing glaciers (e.g. Jakobshavn glacier). Although this remaining low-velocity-variability region includes about 88 percent of the ice sheet's area, it excludes the coastal portions of a few of the catchments, particularly in the southeast where the apparent velocity variability is large.

**2.2 Surface-Mass-Balance and Firn-Density model estimates**

We use our altimetry data to evaluate three state-of-the-art SMB and FD models, that were chosen for this paper because of their low temporal latency, which made them available for the same time period as the recently released ICESat-2 data. Two of these models (MARv3.11.5 and GSFCv1.1) have different surface-mass-balance forcing and a different firn model, but share a similar melt-rate forcing. The third model (GSFCv1.2) updates the surface-melt forcing from GSFCv1.1.

Each model provides estimates of the height change due to surface mass balance and that due to firn-air content change. The sum of these together represent the model estimate of surface-height change due to atmospheric and firn processes. Table S1 gives the internal model variables, and the abbreviations used in this study for each.

**2.2.1 MAR regional climate model**

The Modèle Atmosphérique Regionale (MAR) [*Amory et al.*, 2021; *Fettweis et al.*, 2017; *Tedesco and Fettweis*, 2020]

is a coupled surface-atmosphere regional climate model forced at 6-hourly intervals at the lateral boundaries and ocean surface with climate reanalysis data (here ERA5). It includes detailed snow and firn evolution based on the CROCUS snow

model[*Brun et al.*, 1992; *Brun et al.*, 1989], an atmospheric model [*Gallee and Schayes*, 1994] and a land-surface energy-balance model [*DeRidder and Schayes*, 1997]. MAR has been extensively validated over the Greenland ice sheet, showing general good agreement with weather station and SMB measurements, with some local biases [*Alexander et al.*, 2019; *Fettweis et al.*, 2017; *Fettweis et al.*, 2020; *Montgomery et al.*, 2020]. MAR simulates the top 25 meters of snow, firn and ice, including energy and mass transfer between 30 layers of variable thickness, and incorporates the process of liquid water retention and refreezing. A physically-based scheme is used to simulate snow densification as a function of the weight of overlying snow [*Alexander et al.*, 2019]. Here we use MAR version 3.11.5 (MARv3.11.5) [*Amory et al.*, 2021], which contains modifications to the previous [*Fettweis et al.*, 2020] versions including updates to the cloud parameterization and bare ice albedo adjustments, but without the blowing-snow module. The simulations presented here are run at a spatial resolution of 10 km, forced with the ERA5 reanalysis over 1950-2020 [*Hersbach et al.*, 2020].

### 2.2.2 GSFC modelling based on MERRA-2 and CFM

We generated two sets of SMB and FD products (the GSFC model, after the Goddard Space Flight Center, where the modelling was carried out) using output from a global atmospheric model as input to an open-source FD model. Improvements between the initial (v1.1) and updated (v1.2) versions of this modelling scheme allowed us to explore some of the model processes that can lead to errors in SMB/FD models.

Both versions of the GSFC model used atmospheric variables from the Modern-Era Retrospective analysis for Research and Applications, Version 2 (MERRA-2), developed at the Global Modelling and Assimilation Office (GMAO) at NASA Goddard Space Flight Center [*Gelaro et al.*, 2017]. Atmospheric variables including snowfall, total precipitation, evaporation, 2-meter air temperature, and skin temperature were downscaled to 12.5 km spatial resolution using an offline, high-resolution MERRA-2 replay, in which an atmospheric model (a nonhydrostatic version of the Goddard Earth Observing System Model, version 5 (GEOS-5)) was nudged to match the MERRA-2 reanalysis. In other studies, the improved resolution due to this downscaling technique has led to improved agreement in skin temperature and SMB with other state-of-the-art models over the Greenland ice sheet [*Cullather et al.*, 2014]. One complication in the use of the MERRA-2 model output to drive the firn model was that MERRA-2 does not provide melt as an output. To derive a consistent melt-rate field, we used the MERRA-2 2-m temperatures as input to a degree-day model calibrated to MARv3.5.2 annual melt; the updates between MARv3.5.2 and the MARv3.11.5 model evaluated in this study did not have a major effect on temperature or melt-rate estimates in Greenland, so we assume that the melt-rate calibration for the GSFC models is consistent with MARv3.11.5.

These atmospheric products were used as forcing for the Community Firn Model (CFM; [*Stevens et al.*, 2020]), which provided ice-sheet-wide simulations of the variations in FD through time [*Medley et al.*, 2020]. The configuration of the CFM included several firn processes comprising densification, heat transport, grain-size evolution, meltwater percolation and refreezing, and sublimation. The combination of atmospheric variables from MERRA-2 and output from the CFM produces total firn-column height variations from January, 1980 to December, 2020 at 5-day time steps. The total thickness simulated depends on the ambient climate, typically varying between 118 and 298 m (lower and upper 5th percentile).

Between the GSFCv1.1 and GSFCv1.2 model versions, the positive-degree-day model used to generate melt estimates was refined, and a more complicated model was used to derive the near-surface density [*Medley et al.*, 2022]. For the GSFCv1.1 model, the factor relating the model positive degree days to the melt estimates was derived based on a calibration for each 12.5-km grid cell between the MARv3.5.2 annual melt production and the MERRA-2 2-meter temperature, and the calibration factor for each cell was applied to derive melt estimates from the MERRA-2 temperatures. This calibration yielded calibration factors that were consistent for cells with elevations up to around 1500 m, but increased sharply to large, likely unrealistic values at higher elevation. Thus, to help avoid overestimation of surface melt in the GSFCv1.2 model, any grid cell with a surface elevation higher than 1500 meters was assigned a calibration factor equal to the minimum of the calibrated value and 0.13 kg m$^{-2}$ hr$^{-1}$ K$^{-1}$. This value was chosen based on typical calibration values at 1500 m. The GSFCv1.2 model also included a small increase in the melt-threshold temperature, from the GSFCv1.1 value of 269 K to 270.25 K, which led to minor changes in the calibration factors. This modification was based on an adjustment of model performance as a function of the temperature threshold to better replicate mean melt rates.

For GSFCv1.1, the surface density was assigned based on a linear function of several climatic parameters (wind speed, specific humidity, accumulation rate, and temperature), with coefficients chosen to match the observed surface densities at 151 core sites [*Medley et al,* 2020], while for GSFCv1.2, the surface density was assigned based on a Gaussian Process Regression model relating similar parameters to density, and trained based on a larger set of 187 core sites. This resulted in modestly higher surface densities in GSFCv1.2: The 5-95% range of surface densities for GSFCv1.1 was 247-364 kg m$^{-3}$, while the corresponding range for GSFCv1.2 was 327-387 kg m$^{-3}$ [*Medley et al.*, 2022]. The distribution of density differences between the two models (Fig. S3) is not spatially uniform, with slightly (0-10%) lower densities in GSFCv1.2 in the high-elevation northern portion of the ice sheet, and larger densities in the South and near the coast, although initial evaluations suggest that the low-elevation surface densities in GSFCv1.2 are biased high.

### 2.2.3 Anomaly calculations

For our models, the SMB and temperature data provided do not reflect a steady state climate, and we do not expect the results to converge to any particular equilibrium state. Instead, we choose the period between 1980 and 1995 as a reference period, and calculate the anomalies in surface-height change relative to the mean height change over this period. This is equivalent to assuming, first, that the mean vertical velocity of the ice at the bottom of the firn column is equal to the mean ice-equivalent SMB over this reference epoch, and, second, that any change in the FAC over the reference period reflects a systematic error in the model, whose effects are corrected by subtracting a linear interpolation of the modelled FAC at the beginning and end of the reference period (so that there is no net modelled FAC change during that period), and by extrapolating the same linear relationship to later times. This is consistent with the way in which firn models have been used in correcting altimetry time series (e.g. [*Smith et al,* 2019]), and is useful in this study because it allows us to compare model-predicted changes with less potential influence from long-term drifts in the SMB rate or the total FAC, so that any differences between the models in this study here can only reflect differences relative to the calibration period, with less potential influence

from the spin-up processes used to initialize the models. Although the spin-up of the FD model and our assumption of zero change during the reference period may result in errors in the detrended FD model results (e.g. [*Helsen et al.*, 2008]), we expect these errors to result primarily errors in the modelled height change that are steady over long (decadal) periods of time. The quarter-annual height changes that are the main focus of this study may experience a temporally uniform shift (i.e. might all be too positive or too negative at a particular location) as a result of these errors, but we do not expect the temporal variability of height changes to be significantly affected.

## 2.3 Model-data comparison

For each model, we reduce the full set of 57 million height-difference measurements from ICESAT-2 to a more compact sample with a more even spatial distribution by calculating a block-median set of height differences for each cycle-to-cycle (~91-day) epoch. For each epoch in each model, we assign the height differences into 2.5 km cells for each ICESat-2 pair track. For each such cell in each epoch, we identify the measurement (or measurements) that match (or bracket) the median height difference. For each median difference measurement, we sample each of our model fields (i.e. model total height, model SMB anomaly, model FAC, and model accumulated melt) at the time and location of the height measurements, and calculate their differences. This gives a set of model-field-difference values that are precisely collocated with the measured differences.

### 2.3.1 Weighting the data

After applying the block median to the height differences, there are still substantially more measurements in the northern part of the ice sheet than in the south. Without a correction for this measurement-density bias, differences described by regressions and other block statistics on the residuals would overrepresent the statistics the north of the ice sheet, with less sensitivity to the south of the ice sheet, where some of the most dramatic changes have happened. To help correct for this sampling variability, we calculate the density of measurements (i.e. the number of measurements per square km) on 10-km cells over the ice sheet, and smooth the calculated measurement-density values with a 100-km square averaging kernel. This gives a map of measurement density for the whole ice sheet, which we then interpolate to the difference-measurement locations. We then calculate a weight value for each difference measurement that is equal to the inverse of its interpolated measurement-density value. Because the data gap in June/July 2019 leads to about 50% fewer difference measurements in all epochs that included the second quarter of 2019, we also reduced the weights for all epochs later than Q3-Q4 2019, inclusive, by a factor of two. The resulting weighting ensures that a weighted average of measurements assigns approximately the same weight per unit area to regions in the south of Greenland that it does to regions in the north in addition to accounting for changes in sampling density over time.

### 2.3.2 Regression analysis

To help describe the relationship between modelled and measured height-change estimates, we calculate weighted regressions using components of the models' height changes as independent variables. Our goal in these regressions is to identify how the modelled height changes differ from the measured height differences over the ice sheet. These regressions estimate the scaling(s) for the model parameters that minimize the variance between the measured height differences and the sum of the scaled model parameters:

$$R_{model} = \sum W \left( dh - \left( dh_0 + \sum_{parameter\ j} S_j dP_j \right) \right)^2 \qquad 1$$

Here $W$ are the point-density-based weights from 2.3.1, $dh$ are the measured height differences, $dP_j$ are differences in model parameters interpolated to the locations and times for the measurements that make up each height-change measurement, $S_j$ are the scaling values for each parameter, and $dh_0$ is the mean residual height change. The main statistic we use to evaluate the goodness of fit is the weighted standard deviation, calculated as:

$$\sigma = \left[ \frac{\sum r_i^2 W_i}{\sum W_i} \right]^{1/2} \qquad 2$$

Here $W_i$ are the inverse point densities, and $r_i$ are the regression residuals. As an example, in a regression between the total model height change and the observed height change (section 3.2.1) we solve for the coefficient, $A$, and the mean residual height change, $dh_0$, that minimize the quantity:

$$R_{model} = \sum W(dh - (dh_0 + A\ dh_m))^2 \qquad 3$$

Here $dh_m$ are the modelled height changes. Hypothetically, if one of the models were to systematically overestimate the surface mass balance by a factor of two, we would expect to see a regression for SMB result in a coefficient of 0.5 (meaning that scaling the SMB by 0.5 causes the model to fit the data), and residuals to that regression approximately equal to the data errors. Conversely, if the modelled SMB was not strongly correlated with the measured height differences, we expect to see an arbitrary values for the regression coefficient, and to see a residual variance only slightly smaller than the data variance. Our analysis of the variance statistics is somewhat qualitative because we do not have a convincing way to determine the number of independent parameters in our regressions, but, as will be seen later in this paper, the distinction between variables for which regressions reduce the variance and those for which they do not is usually clear.

To isolate the effects of different processes on the data-model misfit, we calculate regressions against groups of model parameters for a few different spatial and temporal subsets of the data. Because regressions can be sensitive to points with large residuals that are not representative of the statistics of the data, for each subset of the data, we first remove large outlying difference values caused by, among other things, complex and steeply sloping ice surfaces, and blunders in the ATL06 data underlying the ATL11 data. To identify these, we calculate the robust spread of height-difference distribution (here defined as

the half width of the central 68% of the distribution) and remove from the analysis any outlying points whose difference values
are more than 12 times this spread away from the mean. This editing strategy is applied iteratively until subsequent means are
identical or until ten iterations are complete. The final regressions and their residual are calculated after these outlying points
are removed. For the ice sheet as a whole, this editing procedure removes about 1% of points, of which the standard deviation
is equal to 2.3 m.

We perform regressions for the total model change ($dh_m$), the height change due to SMB anomalies ($dh_{SMB}$), and the
height change due to firn-air content ($dh_{FAC}$). We use the modelled total melt to segregate the data into strong-melt ($z_{melt} > 0.2$
$|dh_m|$) and weak-melt ($z_{melt} < 0.2$ $|dh_m|$) subsets, but do not perform explicit regressions between surface-mass-balance change
and melt because melt does not have a consistent linear relationship with surface-height change. The height change associated
with melt depends on the density of the snow or ice being melted, and on whether the meltwater runs off the ice sheet or is
refrozen, which makes the results of a regression between $z_{melt}$ and $dh$ more difficult to interpret than those for the other
variables.

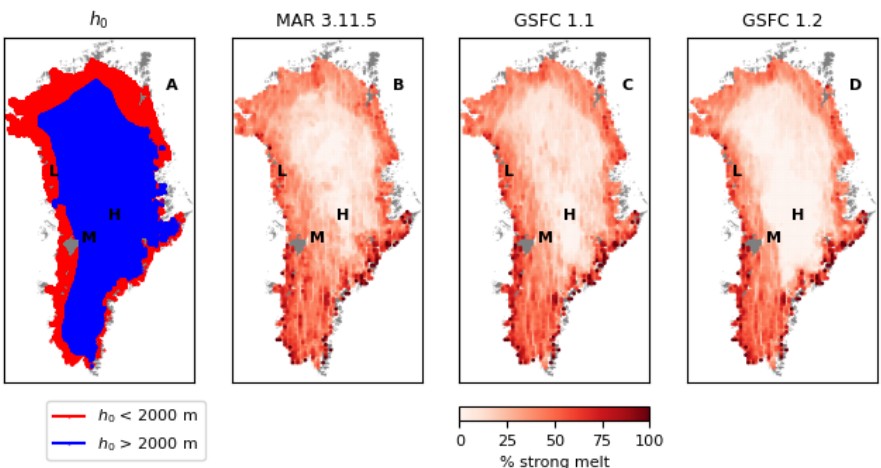

**Figure 1: Spatial distribution of elevation and melt categories. Panel A shows the location of the low-and high-elevation data. Panels B, C, and D show the fraction of all points in 10x10-km bins for each model that fall into the strong-melt ($z_{melt} > 0.2$ $|dh_m|$) category.**
**Letters 'H', 'M', and 'L' indicate locations for plots in Fig.2.**

### 2.3.3 Model Subsets

To help identify the processes at work in determining the model-data misfit, we divided the data into low- and high-
elevation subsets (h<2000 m and h>2000 m), into weak and strong-melt subsets using the models' accumulated melt parameter,
$z_{melt}$ ($z_{melt} < 0.2|dh_m|$, $z_{melt} > 0.2|dh_m|$), and isolated the high-melt time periods for which we have data from both 2019 and 2020
(spring-summer, spanning Q1-Q2 and Q2-Q3 for each year). Figure 1 shows maps of the elevation subsets, and of the spatial

distribution of high-melt differences. Based on Fig. 1, we can see that all three models estimate frequent strong melt in low-elevation regions around the coast, and that both MARv3.11.5 and in GSFCv1.1 estimate that melt occurred sporadically throughout the ice sheet, while GSFCv1.2 estimates that strong melt was rare in the high-elevation portion of northern Greenland.

Table 2 gives some general properties of the model outputs for all of the data together, and for each subset. Based on these values, we can see that melt was considerably stronger in 2019 than it was in 2020 for all three models, with nearly three times as much of the data weight falling into the strong-melt category (compare the f_melt statistics for sp-su 2019 with those from sp-su 2020). The RMS statistics of the SMB variables reflect the strong surface-melt signal in the summer of 2019, and the large melt signals associated with the lowest-elevation part of the ice sheet. FAC variability is largest in 2019, and for

both strong-melt subsets of the data. The fraction-of-weight column indicates that the largest fraction of the data (by weight) fell into the high-elevation, weak-melt category, while the smallest fraction fell into the low-elevation, strong-melt category. MARv3.11.5 and GSFCv1.1 had similar distributions of data weight and variance among the subsamples, but GSFCv1.2 had less weight in the strong-melt category, particularly in the high-elevation region, and had smaller FAC variance within the high-melt subsamples. Note that because of the outlier editing applied to each subset, the superset of the high- and low-

elevation, weak- and strong-melt subsets contain a slightly larger (~1.3%) total weight than does the 'all' subset for each model. We do not believe that including or omitting these points makes a large difference in our results.

| Model | Elev. | subset | f_wt | f_melt | sigma_SMB | sigma_FAC | RMS_data |
|---|---|---|---|---|---|---|---|
| MARv3.11.5 | all | all | 1.00 | 0.29 | 0.17 | 0.24 | 0.25 |
| | all | sp-su 2019 | 0.22 | 0.60 | 0.34 | 0.42 | 0.43 |
| | all | sp-su 2020 | 0.28 | 0.34 | 0.16 | 0.22 | 0.22 |
| | high | weak melt | 0.50 | 0.00 | 0.04 | 0.09 | 0.12 |
| | low | weak melt | 0.22 | 0.00 | 0.13 | 0.18 | 0.26 |
| | high | strong melt | 0.15 | 1.00 | 0.08 | 0.39 | 0.24 |
| | low | strong melt | 0.15 | 1.00 | 0.59 | 0.49 | 0.74 |
| | | | | | | | |
| GSFC v1.1 | all | all | 1.00 | 0.28 | 0.14 | 0.20 | 0.25 |
| | all | sp-su 2019 | 0.22 | 0.64 | 0.26 | 0.34 | 0.43 |
| | all | sp-su 2020 | 0.28 | 0.29 | 0.13 | 0.17 | 0.22 |
| | high | weak melt | 0.50 | 0.00 | 0.04 | 0.08 | 0.12 |
| | low | weak melt | 0.21 | 0.00 | 0.15 | 0.17 | 0.26 |
| | high | strong melt | 0.14 | 1.00 | 0.09 | 0.35 | 0.24 |
| | low | strong melt | 0.15 | 1.00 | 0.44 | 0.36 | 0.73 |

| | | | f_wt | f_melt | | | |
|---|---|---|---|---|---|---|---|
| GSFC v1.2 | all | all | 1.00 | 0.23 | 0.14 | 0.13 | 0.25 |
| | all | sp-su 2019 | 0.22 | 0.49 | 0.27 | 0.22 | 0.43 |
| | all | sp-su 2020 | 0.28 | 0.24 | 0.12 | 0.11 | 0.22 |
| | high | weak melt | 0.55 | 0.00 | 0.04 | 0.06 | 0.12 |
| | low | weak melt | 0.22 | 0.00 | 0.14 | 0.11 | 0.26 |
| | high | strong melt | 0.10 | 1.00 | 0.09 | 0.23 | 0.29 |
| | low | strong melt | 0.14 | 1.00 | 0.47 | 0.26 | 0.75 |

**Table 2. Properties of subsamples of the models. The elevation and subset columns indicate the subsample of data for which the statistics were calculated. The elevation column indicates 'low' for elevations below 2000 m, 'high' for elevations above 2000 m, or 'all'. The subset column indicates 'strong melt' for data for which $z_{melt} > 0.2 \, |dh_m|$, 'weak melt' for data for which $z_{melt} < 0.2 \, |dh_m|$, or indicates temporal divisions of the data. The 'f_wt' column indicates the fraction of the total data weight for each subsample of the data. The f_melt column indicates the fraction of points in each subsample that fell into the strong-melt category.**

## 3. Results

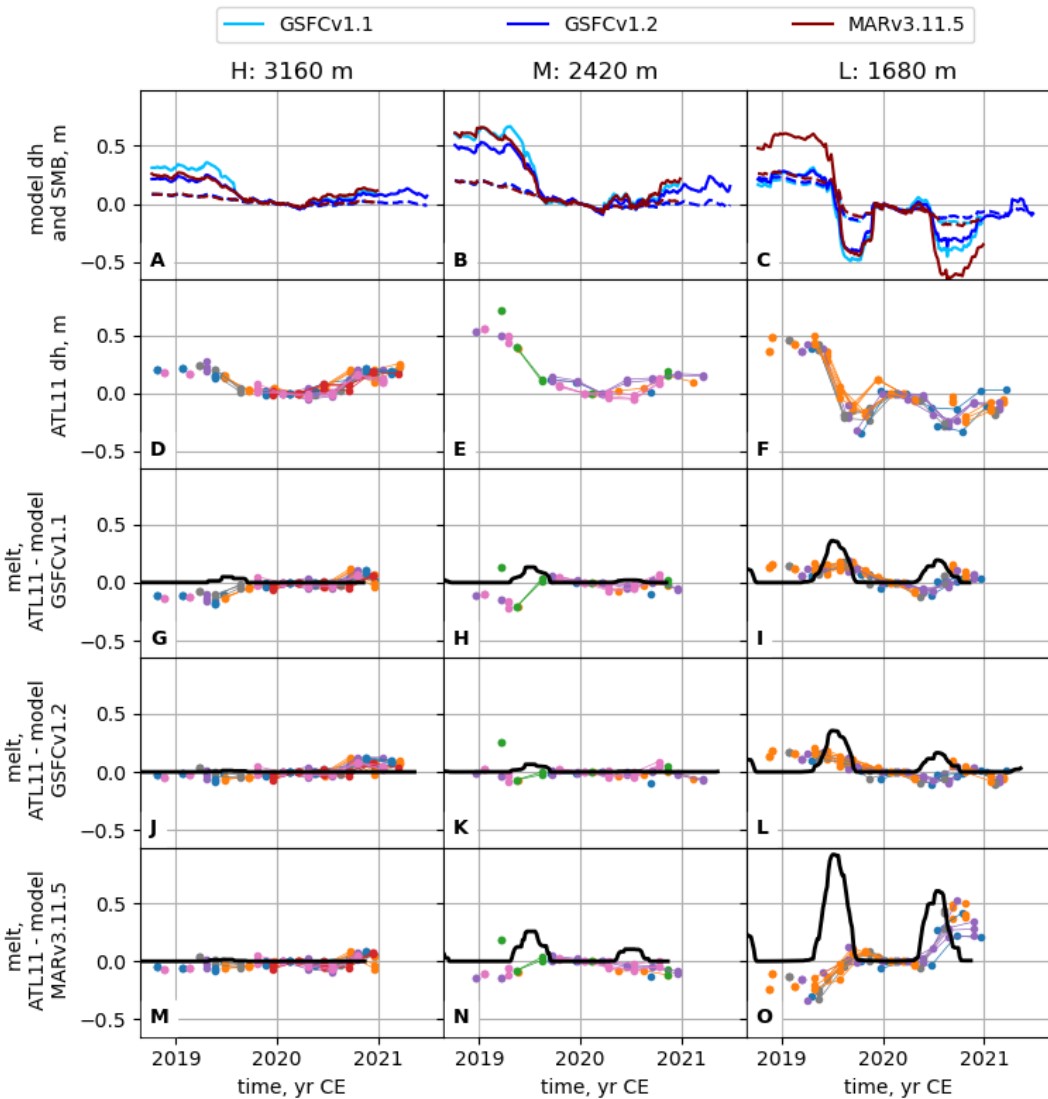

**Figure 2. Data and Model estimates for three locations indicated in Fig. 1. A-C: Model total height anomalies (solid) and SMB-driven height anomalies (dashed) for all the three models. D-F: ATL11 height time series for each location, with different colours indicating different RPTs. G-O: Differences between ATL11 time series and total model height anomaly for each model and location. Solid black lines show a 90-day running mean of the melt estimated by each model (positive indicates height lost to melt). To align the time series vertically, we subtract the value for cycle 6 (Q1 of 2020) from each. Time series for each RPT are joined by a solid line when derived from continuous repeat-track measurements. Broken lines or lone points indicate crossover measurements or missing values in the repeat-track measurements.**

Figure 2 shows height-change measurements, model data, and measurement-model residuals for three 20x20 square regions in Greenland with different melt characteristics (locations indicated in Fig. 2). For each region, we plot the SMB anomalies converted to height (dashed lines) and the total height anomalies (solid lines) for each model (A-C). We plot and the mean

ATL11 height differences for each RPT within each box (D-F, different colours for different RPTs), as well as the residuals between the ATL11 height differences and predicted height change for GSFCv1.1 (G-I), GSFCv1.2 (H-J), and MARv3.11.5 (K-M), and the models' melt estimates. The three locations show a range of melt intensities: At the high-elevation location, (3160 m, 'H' in fig. 2), only the GSFCv1.1 shows a visible melt signal, while at the middle-elevation location (2420 m, 'M'), GSFC1.1 and MARv3.11.5 each have decimeter-level melt signals in the summer of 2019, while GSFCv1.2 has only a few

cm of melt. At the lower-elevation location (1680 m ,'L') all three models have tens of cm of melt in the summers of 2019 and 2020. For the high-elevation location, the ATL11 data show a dip in elevation over the summer of 2019, recovered the following winter. All three models match the data well, except that GSFCv1.1 overpredicts the summer-2019 drawdown by around 10 cm. For the middle-elevation location, the summer drawdown is larger, and is predicted well by GSFCv1.2, and overpredicted by the other two models. At the lower-elevation location, the ATL11 data show a drops of 20-80 cm during the

summers, which are matched well by the GSFC models, and overpredicted by MARv3.11.5. These examples span much of the range of variability seen around the ice sheet but are not necessarily typical of any larger subset of the data, particularly with respect to the relative accuracy of GSFCv1.1 and MARv3.11.5, which can be notably different between measurements separated by a few tens of km, likely because of the small-scale variability in melt shown for these models in figure 1. The relative variability of the models' surface-height and SMB fields seen in these examples, however, are consistently seen

throughout the ice sheet: Over much of the area covered by our masked data, runoff plays only a small role in SMB variations, which limits the temporal variability in the SMB height-change estimates, so that SMB variations consist of small height increases driven by accumulation events, balanced by the detrending in the anomaly calculations. The surface-height predictions, often have much larger seasonal cycles that reflect changes in FAC, because melt events result in a loss of low-density snow, where the meltwater infiltrates deeper layers to fill pore space.

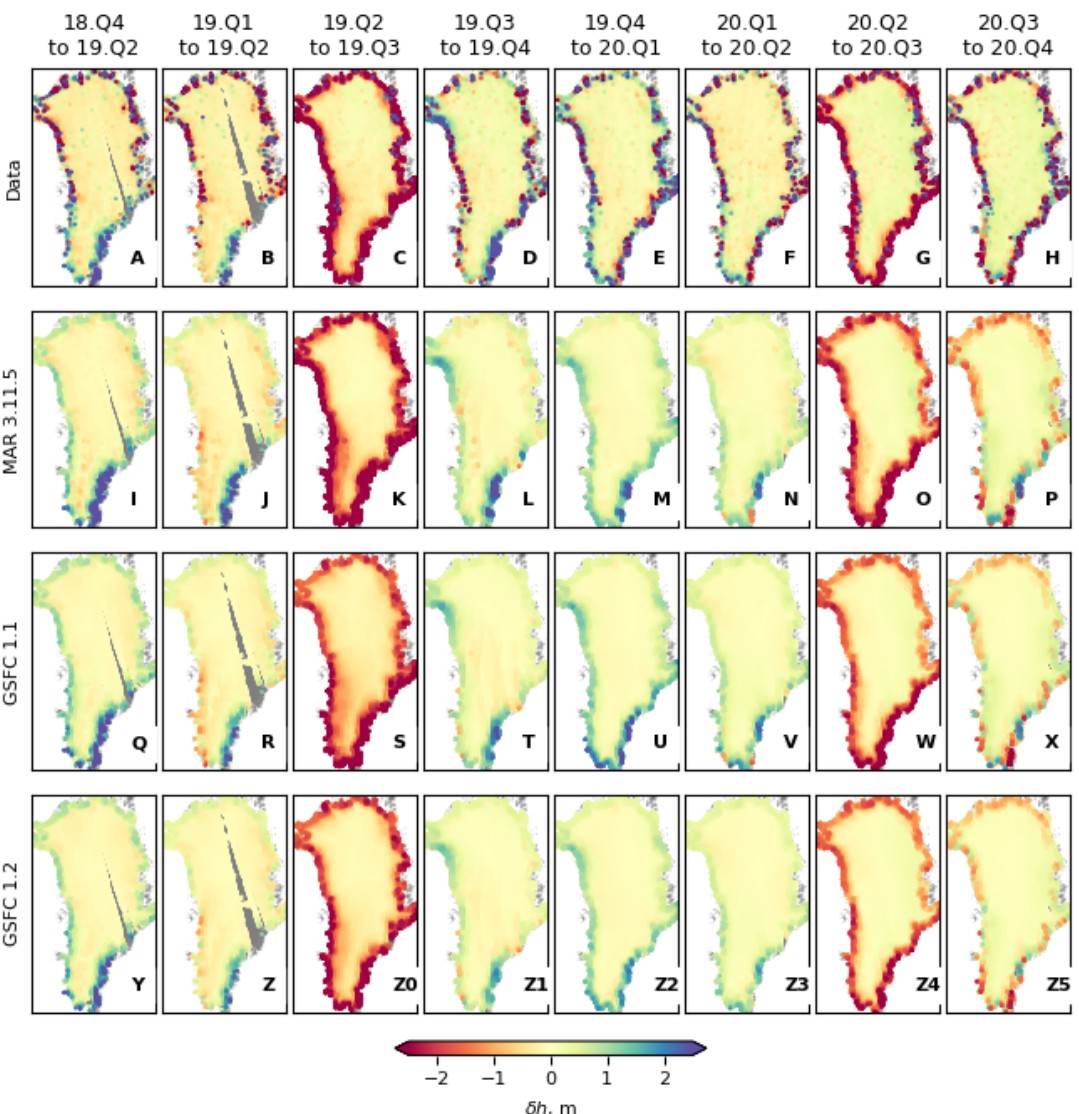

**Figure 3: Measured height differences in Greenland from ATL11 (A-H), and the predicted height changes from MAR3.11.5 (I-P), from GSFCv1.1 (Q-X), and GSFCv1.2 (Y-Z5). Points on the plots show the measured or predicted height differences for each epoch (listed at top), and are masked to include only on-ice points, but are not masked based on velocity variations.**

## 3.1 Measured changes and model residuals

Figure 3 (panels A-H) shows maps of height differences ($\delta h$) measured by ICESat-2 between October 2018 and December 2020. This span includes both the 2019 and 2020 melt seasons (2019 Q2-Q3 and 2020 Q2-Q3), of which 2019 appears to have been substantially more intense. The 2019 melt season shows thinning over most of the ice sheet, with height lost even in the middle of the ice sheet where melt is typically rare. The strongest thinning rates were concentrated within a few tens of kilometers near the coast, declining inland, with the largest inland extent of thinning in the southwest and with

relatively small thinning extent in the northeast. The summer of 2020 shows a narrower band of thinning, confined mostly to the coast. The fall and winter seasons show narrow coastal bands of thickening reflecting winter snowfall, with the largest values in the Southeast, where the maximum precipitation rates occur [*Fettweis et al.*, 2020].

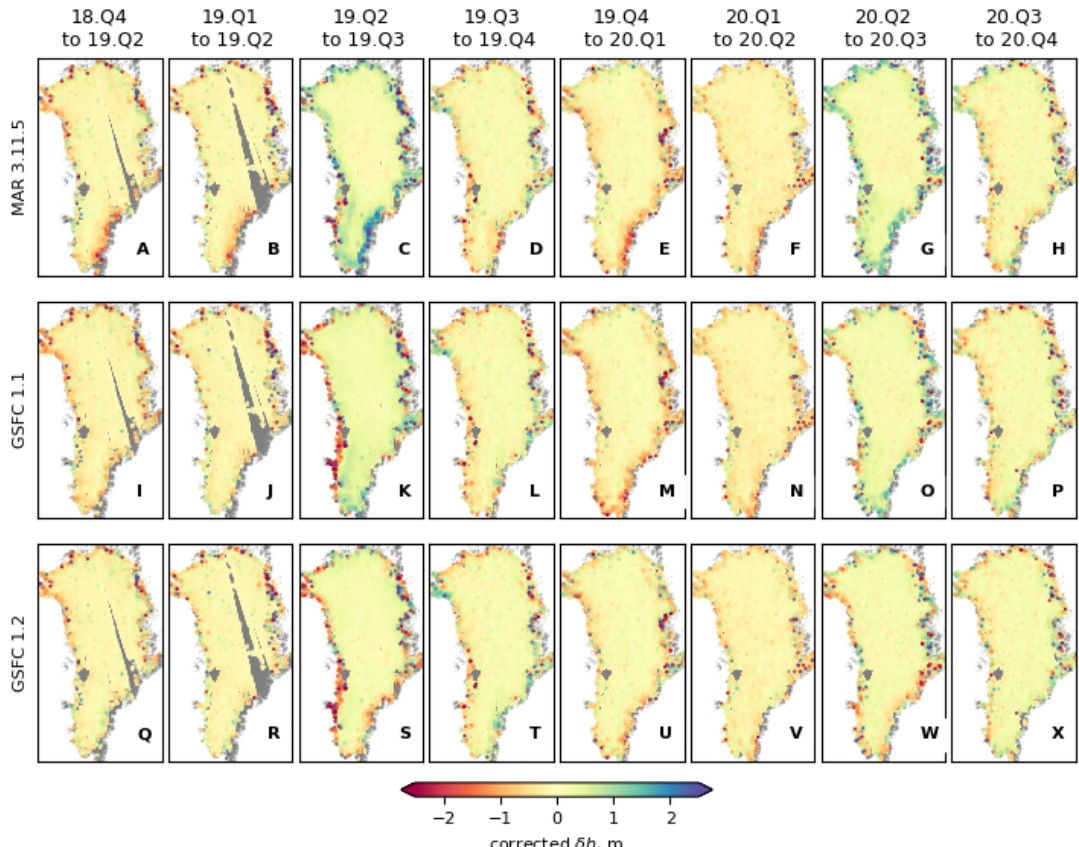

**Figure 4: Residuals between measured and predicted height changes from MAR3.11.5 (A-H), GSFCv1.1 (I-P) and GSFCv1.2 (Q-X). Measurements are masked to remove areas where there have been large velocity variations.**

Fig. 3 (panels I-Z5) show the corresponding height differences expected solely due to modelled SMB/FD processes for each epoch ($\delta h_m$). These plots show that the models match the overall temporal and spatial extent of the thinning in the summers of 2019 and 2020, but the predicted thinning is somewhat more spatially extensive than the observed thinning; likewise, winter and fall thickening appears to be more spatially extensive in the models than in the measured values.

Figure 4 shows the residuals between the measured height differences and the changes predicted by the models, which we term the 'corrected' height changes ($\delta h_c$). Consistent with the time series in Fig. 2, these plots show that $\delta h_c$ has smaller variations than $\delta h$ over most of the ice sheet, but that in many parts of the ice sheet, subtracting the predicted SMB/FD signal reverses the sign of the observed differences. This indicates that the models predict larger changes than are observed in the data: For example, MARv3.11.5 and GSFCv1.1 predict stronger thinning between Q2 and Q3 of 2019 than was measured

resulting in positive corrected values (Fig. 4C, 4K), and these overestimates of thinning are mirrored by overestimates of

thickening during the colder seasons (Fig. 4E, 4M). During the summer of 2019, the GSFC models appear to underestimate thinning in the low-elevation bare-ice zone of the southwest margin (Fig. 4K, 4S), while MARv3.11.5 appears to be much closer to correct (Fig. 4C).

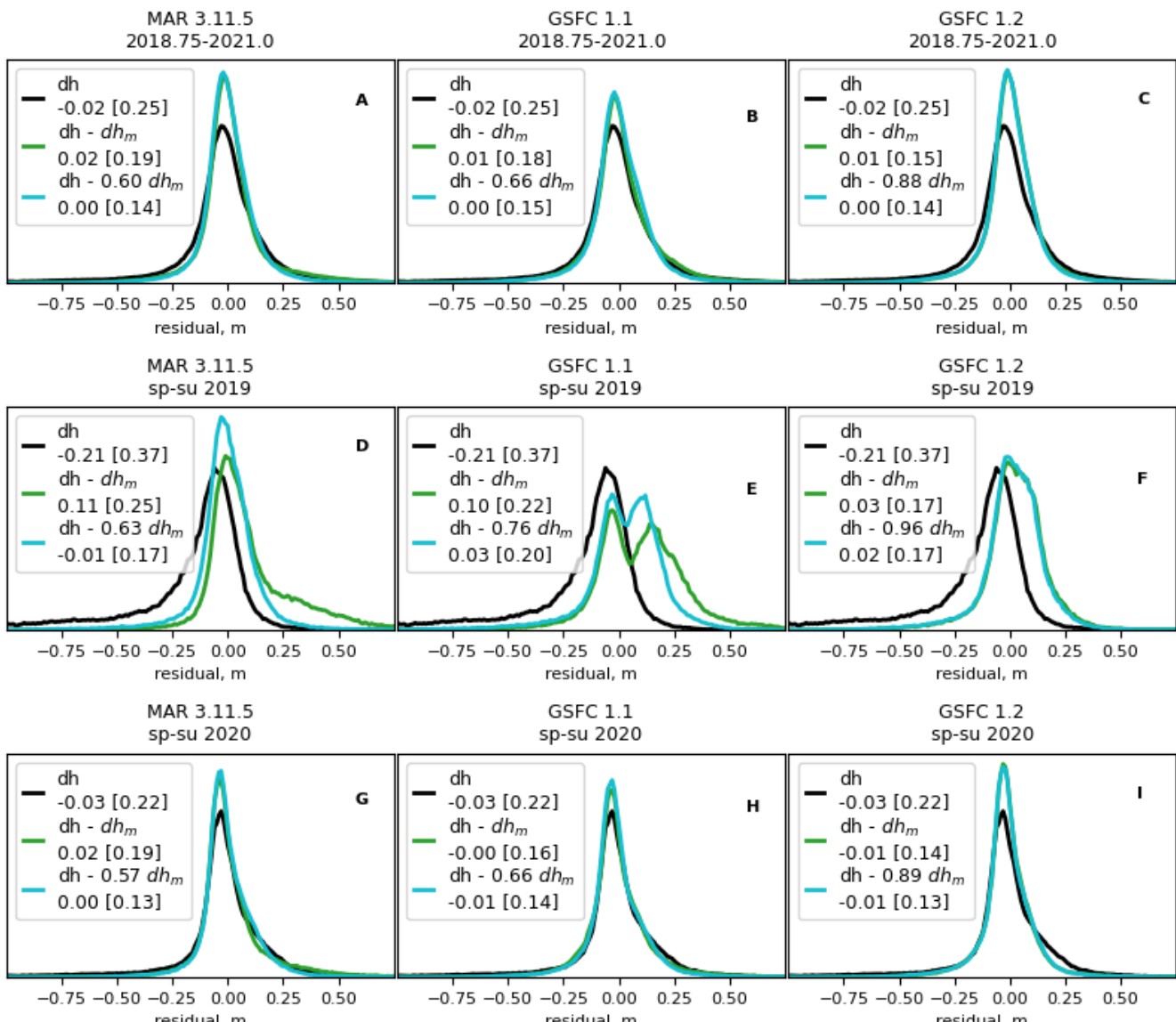

Figure 5: Height-difference and residual histograms for the whole ice sheet, for three time periods. The curves in each plot represent (black, dh): measured height change; (green, dh-dh$_m$): height change corrected by the total model; (blue, dh- [A] dh$_m$): height change, corrected by model scaled by [A]. Vertical axis units are arbitrary, but the areas under the curves within each plot are the same. Plots in the top row (A-C) shows measured and model-residual distributions for all epochs together (Q3 2018- Q4 2020). Plots in the middle row (D-F) show distributions for spring-summer for 2019 (Q1-Q2 and Q2-Q3), the bottom row shows corresponding distributions for 2020. In the captions to each histogram, *dh* indicates the measured height differences, *dh$_m$* indicates the modelled height changes. The second row of each caption gives the mean and [standard deviation] of the distribution in each histogram.

## 3.2 Regression experiments

### 3.2.1 Regression of height change against total model change, for temporal subsamples of the data

Figure 5 shows histograms of observed height differences, model-corrected height differences, and regression residuals for the full time series, and for subsamples of the data spanning the spring-to-summer and summer-to-autumn epochs of 2019 and 2020. For each subset of the data, we plot the histograms of the data (measured height differences), of the residuals between the unscaled model and the data (equivalent to the residuals for A=1 in Eqn. 1) and of the residuals between the scaled model and the data (after solving for the value of $A$ that minimizes $R_{model}$). The legend for each panel in figure 4 specifies the

quantity shown in the histogram, the mean residual (equivalent to $dh_0$ in Eqn. 1) and the scaled standard deviation (Eqn. 2).

Distributions of measured height differences (Fig. 5, A-C) have a standard deviation of 0.25 m and a mean of -0.02 m, with a tail of negative values likely corresponding to the strong low-elevation melt-season drawdown. Evaluating the corrections and regressions shows a clear distinction between two of the models (MARv3.11.5 and GSFCv1.1) and the third (GSFCv1.2); we describe the results from the first two, and then describe how these differ from those of the third. Subtracting

the models gives distributions with smaller standard deviations (0.19 m for MARv3.11.5, 0.18 m for GSFCv1.1) and near-zero means. Results from the regressions show that when the MARv3.11.5 and GSFCv1.1 are rescaled by 0.63 and 0.76, respectively, the residual standard deviations are notably smaller (0.14 m for MARv3.11.5, 0.15 for GSFCv1.1), and the residual means remain close to zero. For these models, the unscaled model outputs account for less than half the variance in the measured height differences (39% for MARv3.11.5, 46% for GSFCv1.1), while the scaled models each account for more

than 60% of the variance (see tables S2-S4 for variance statistics). By contrast to MARv3.11.5 and GSFCv1.1, the residuals to the *unscaled* GSFCv1.2 model are smaller (at 0.15 m), the optimal rescaling is close to unity (0.88), and the residuals to the rescaled model (at 0.14 m) are only marginally smaller than those of the unscaled model.

The data from spring-summer 2019 (Fig 5, D-F) show substantial (-0.21 m), average ice-sheet height loss, and larger (0.37 m) height-difference standard deviations. Similar to the full time series, the residuals to the unscaled models have smaller

spreads than do the data (0.25 m for MARv3.11.5, 0.22 m for GSFCv1.1), and the spreads of the residuals to the optimally rescaled models (by 0.63 for MARv3.11.5, 0.76 for GSFCv1.1) are yet smaller (0.17 m for MARv3.11.5, 0.20 m for GSFCv1.1). For both MARv3.11.5 and GSFCv1.1, the mean residuals to the unscaled models are positive, (around 0.10 m for both), and the mean residual to the scaled models are close to zero (-0.01 m for MARv3.11.5, 0.03 for GSFCv1.1). For MAR, the summer-2019 unscaled-residual histograms (Fig. 5D) show a positive tail of residuals that mirrors the negative tail seen in the

uncorrected data. For GSFCv1.1, instead of a large positive tail, the unscaled-residual histograms show a second peak at around 0.25 m, suggesting that for some points where MARv3.11.5 overcorrects the data, GSFCv1.1 overcorrects the data less drastically. Rescaling the models reduces the magnitude of the positive tail of residuals for MARv3.11.5 and brings the second peak for GSFCv1.1 closer to zero. As we observed with the full time series, in contrast to the other two models, GSFCv1.2

has an optimal 2019 rescaling coefficient close to unity (0.96), and the unscaled and scaled models make substantial reductions
in the standard deviation (to 0.17 m, comparable to the MARv3.11.5 rescaled model).

The spring-summer-2020 statistics and histograms are similar to the full-time-series statistics, and the rescaling coefficients and the improvements in residuals due to the rescaling are identical within a few percent to those of the full time series.

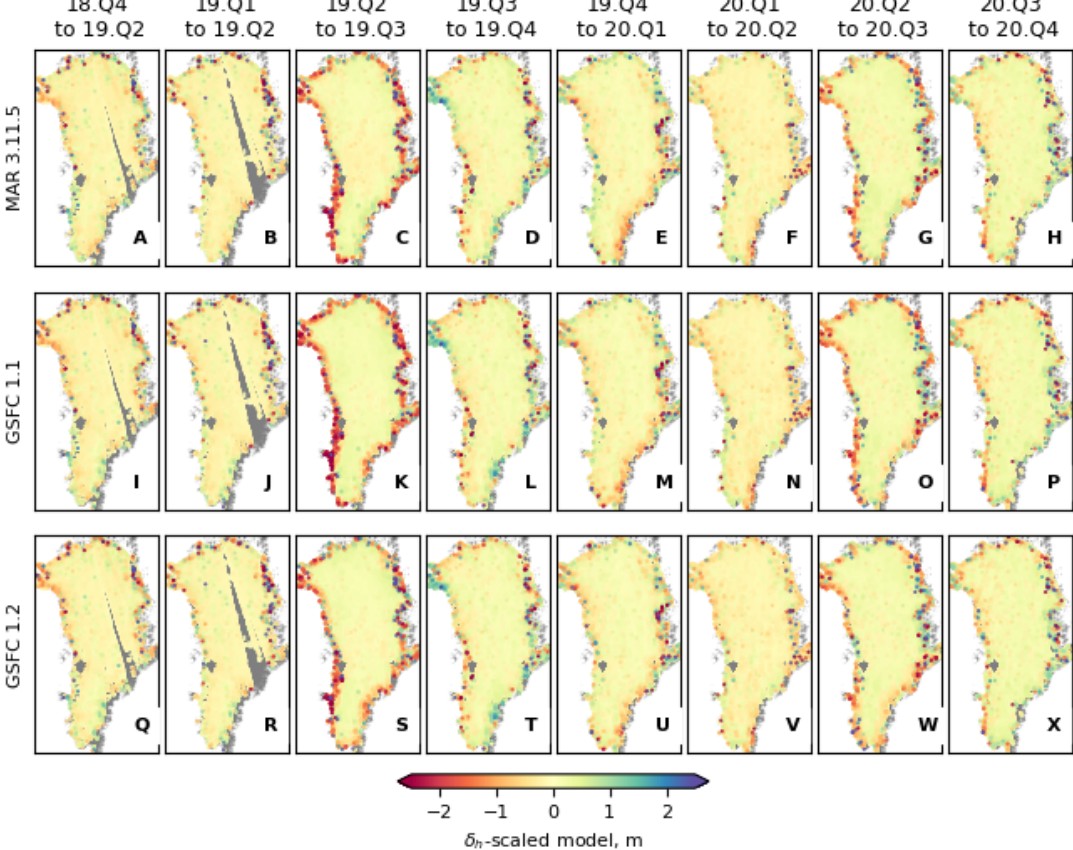

**Figure 6: Residuals between measured height differences and rescaled MARv3.11.5 (A-H), GSFCv1.1(I-P) and GSFCv1.2 (Q-X) models. The time periods for each set of differences is indicated at the top of each column. MARv3.5.11 is rescaled by 0.61, GSFCv1.1 is rescaled by 0.66, and GSFCv1.2 by 0.88. In each plot, the data are masked to remove areas with large velocity variations.**

Maps of the residuals to the rescaled MARv3.11.5 and GSFCv1.1 models (Figure 6, A-P) show that the model overcorrections that were apparent as a blue-tinged rim around the ice sheet in the summer epochs in Fig. 4 (panels C, K, G, and O) are much less prominent, although for some points immediately adjacent to the margin during the summers, the rescaled model under corrects for the measured height differences, resulting in locally larger residuals. For GSFCv1.2, the maps of residuals to the rescaled model (Fig. 6, Q-X) are not visibly different to those of the unscaled model and are visually similar to the rescaled residuals from the other two models.

### 3.2.2 Regression of height change against total model change, for melt and elevation subsamples of the data

The behaviour of the ice sheet and the models was evidently substantially different between the spring-summer subsample of 2019 and the rest of the model domain. To explore the role of melt in the data-model differences, we subdivide the full time series of data into four groups of difference measurements based on model melt and elevation: One division splits the data between strong-melt ($z_{melt} > 0.2$ |dh$_m$|) and weak-melt ($z_{melt} < 0.2$ |dh$_m$|) differences, the other between low-elevation ($<$ 2 km) and high-elevation ($> 2$ km) differences. Figure 7 shows regression results for these four subsamples of the data, for

the entire model time span. Note that in these plots, the points included in the histograms are slightly different from model to model, because the points included in the strong- and weak-melt subsamples were determined by each model's melt field.

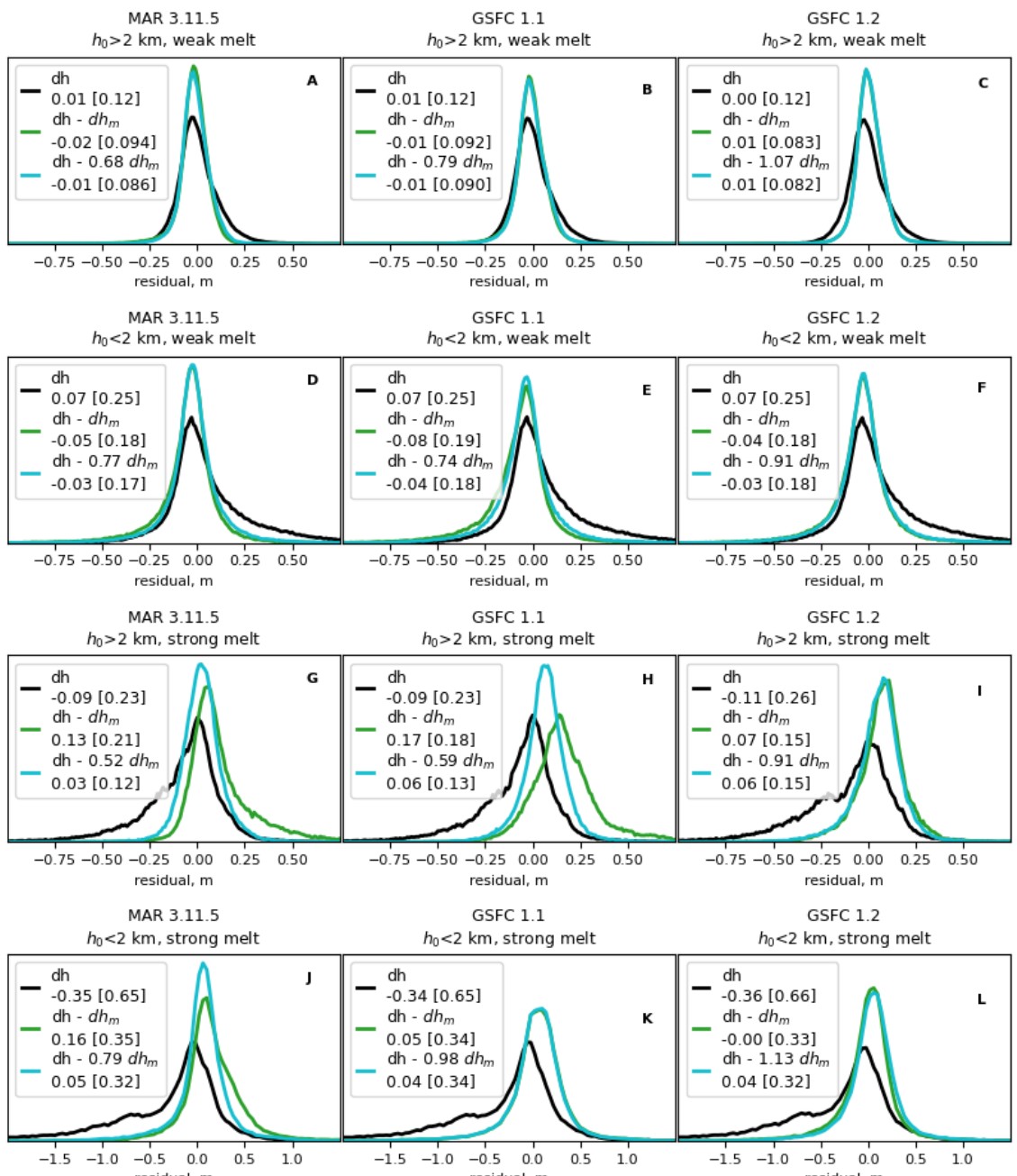

**Figure 7: Height-difference and residual histograms for subsamples of the data based on height and model estimates of melt and SMB.** The curves in each plot represent (black, dh): measured height change; (green, dh-dh$_m$): height change corrected by the total model; (blue, dh- [A] dh$_m$): height change, corrected by model scaled by [A]. Vertical axis units are arbitrary, but the areas under the curves in each panel are the same. Panels A-C show data from high elevation (h$_0$> 2km) with weak melt (z$_{melt}$< 0.2 |dh$_m$|) for the three models. Panels D-F show data from low elevation (h$_0$> 2km) with weak melt. Panels G-I show data from high elevation (h$_0$> 2km) with strong melt (z$_{melt}$< 0.2 |dh$_m$|). Panels J-L show low elevation with strong melt. Legend captions follow the format used in Fig. 5.

For the weak-melt subsamples of the data (Fig. 7, A-F), the MARv3.11.5 and GSFCv1.1 models perform well with no rescaling, reducing the high-elevation residuals from 0.12 m to around 0.09 m, and the low-elevation residuals from 0.25 m to ~0.18 m. The optimal rescalings (between 0.68 and 0.79) result in marginally smaller residuals.

For the strong-melt, high-elevation subsample of the data (Fig. 7, G-I) the results are markedly different. The data in
these subsamples show substantial mean height loss (~0.1 m) and spreads of 0.23-25 m. Corrections for the unscaled MARv3.11.5 and GSFCv1.1 models result in small-to-moderate reductions in the residual spread. In the case of MARv3.11.5, the resulting histogram has positive peak, and a strong tail of values in the positive direction, and in the case of GSFCv1.1, the histogram has a peak substantially shifted in the positive direction, but a smaller tail of positive values. Comparisons to D and E in Fig. 5 suggest that high-elevation melt events account for the shapes of the unscaled-model residual histograms for the
spring-summer 2019 period. For both MARv3.11.5 and GSFCv1.1, rescaling the model by just more than half resulted in a substantial decrease in the residual standard deviation, to 0.12 m and 0.13 m, respectively. This is in contrast to the GSFCv1.2 model, for which the optimal scaling is close to unity (0.91) and the unscaled model has residuals that are comparable in magnitude to those of the rescaled MARv3.11.5 and GSFCv1.1 models.

For the strong-melt, low-elevation subsample of the data (Fig. 7, J-L), the histograms of uncorrected height
differences have a near-zero peak, with a large negative tail of values indicating strong summer drawdown, a substantial negative mean, and standard deviations of around 0.65 m. All three models correct for a large fraction (71-75%) of the variance in the data, all have optimal rescaling values close to unity (between 0.79 for MAR3.11.5 and 1.13 for GSFCv1.2), which make very small improvements in the residuals over the unscaled models.

### 3.2.3 Regression of height change against model components, for melt and elevation subsamples of the data

To further explore the importance of different components of modelled height change, we perform regressions in which we allow different scaling factors for individual components of the SMB and FD models. In these experiments, we solve for the coefficients, B, and C that minimize the quantity:

$$R_{comp} = \sum W\big(\boldsymbol{dh} - (dh_0 + B\,\boldsymbol{dh_{SMB}} + C\,\boldsymbol{dh_{FAC}})\big)^2 \qquad\qquad 4$$

Here $\boldsymbol{dh_{SMB}}$ is the modelled height change due to SMB variations, and $\boldsymbol{dh_{FAC}}$ is the modelled height change due to FAC
variations. For each subset of the data, we perform three sets of regressions: in the first, we set B=1, to solve only for the scaling of $dh_{FAC}$, in the second we set C=1, to solve for the optimal scaling of $dh_{SMB}$, and in the third, we solve for B and C together. Because rescaling the components individually for the weak-melt data changes the statistics of the residuals only marginally, we do not present these histograms, and we likewise omit the GSFCv1.2 model. Instead, Figure 8 shows histograms for the optimal scaling of the FAC and SMB components for the MARv3.11.5 and GSFCv1.1 models, for both
elevation subsamples of the strong-melt data. For each model and elevation category, we show histograms of the observations

corrected with the unscaled model, the observations corrected for each height change component scaled separately (i.e. with the full correction applied for the other component), and for the two components scaled independently.

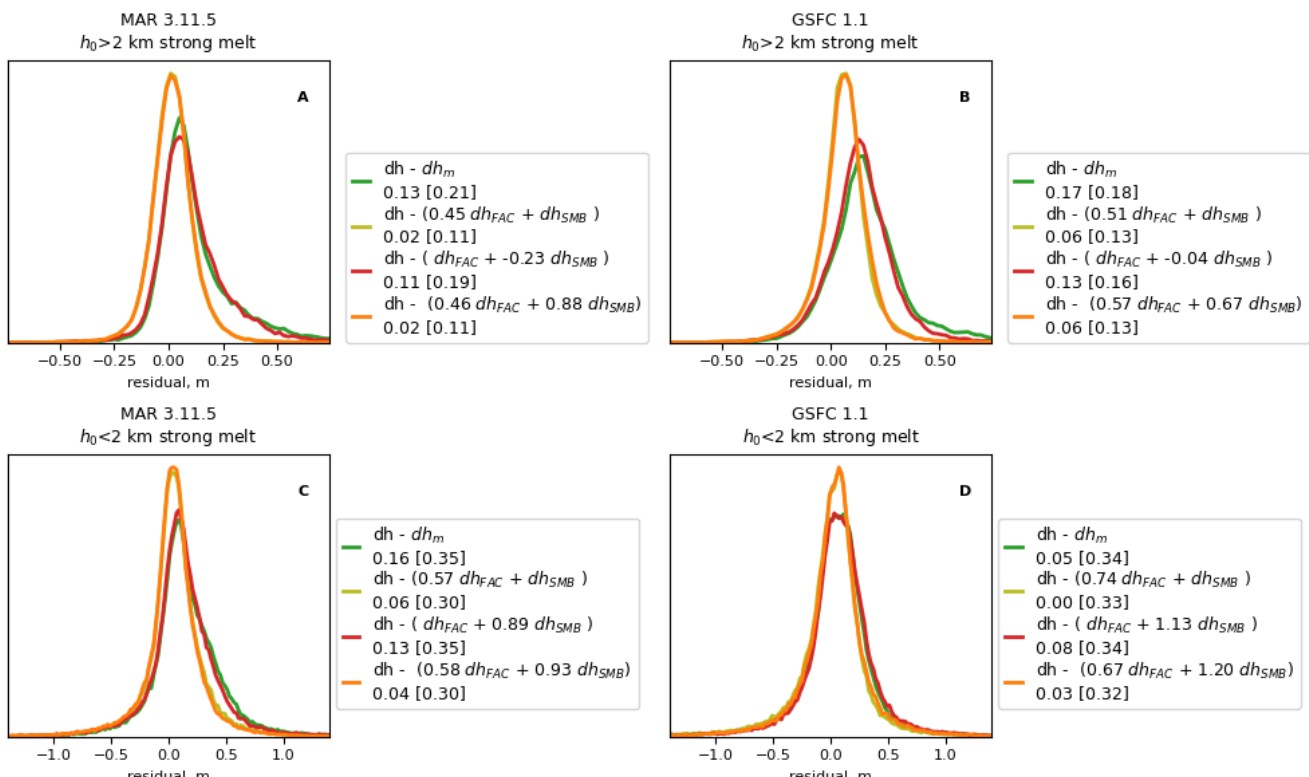

Figure 8. Unscaled-model residuals and regression residuals for individual model components of the MARv3.11.5 (A, C) and GSFCv1.1 (B, D) models, for the high-elevation (A, B) and low-elevation (C, D) subsamples of the high-melt ($z_{melt} > 0.2 |dh_m|$) data. Model components listed in the captions are the firn-air-column change ($dh_{FAC}$) and the surface-mass-balance height change ($dh_{SMB}$). The mean and [standard deviation] of the residuals are given in each caption. Note that in each of these plots, the histogram for the separate rescaling of $dh_{FAC}$ and $dh_{SMB}$ is nearly identical to that for rescaling of $dh_{FAC}$ alone, so the histograms for $dh_{FAC}$ are not separately visible.

For the high-elevation data and MARv3.11.5 (Fig. 8A), rescaling of the FAC makes a notable reduction in residuals relative to the unscaled model, with only small improvements due to rescaling the SMB. The combination of the FAC rescaled by 0.45 and the full SMB leaves a residual standard deviation of 0.11, while the SMB rescaled (by -0.23) plus the unscaled FAC leaves a larger (0.19 m) standard deviation. The optimal scaling for the two together, of 0.46 for FAC and 0.88 for SMB, produces results essentially identical to the rescaling of the FAC alone (0.11 m standard deviation).

We see similar results for the high-elevation data and the GSFCv1.1 model (Fig. 8B), where the combination of the FAC rescaled by 0.51 and the full SMB correction leaves residuals with a standard deviation of 0.13 m, approximately the same as seen for rescaling of the complete model (Fig. 7H). The optimal scaling for the SMB alone is -0.04, so rescaling for the SMB alone is essentially equivalent to using the unscaled FAC alone as a correction; this leaves a residual standard

deviation of 0.16 m. Adjusting both SMB and FAC yields optimal scaling factors of 0.57 for FAC and 0.67 for SMB, but

approximately the same standard deviation as the FAC scaling alone.

For the low-elevation, high-melt subsample, none of the rescalings result in large reductions in the residual standard deviations. The largest reduction is for MARv3.11.5 (Fig. 8C), where the FAC scaled by 0.57 plus the unscaled SMB yields a residual standard deviation of 0.30 (which should be compared to 0.35 for the unscaled model), and scaling the SMB (by an optimum value of 0.89) makes little or no improvement over the unscaled model. For GSFCv1.1 (Fig. 8D), the high-melt,

low-elevation regressions all produce residual standard deviations at most 1-2 cm smaller than those from the unscaled model.

For the GSFCv1.2 model, the optimal rescalings do not make any notable improvement in the residuals over the unscaled model for either subsample of the data (see Supplemental Material figure S4). The spread of residuals to the unscaled model for the high-elevation subsample of the data (0.15 m) is comparable to that of the fully rescaled GSFCv1.1 (0.13 m), but slightly larger than that of the rescaled MARv3.11.5 (0.11 m). For the low-elevation subsample, the GSFCv1.2 model,

like the other two, has a residual spread of around 0.33 m for the unscaled model, and none of the rescalings improves the spread by more than 0.01 m.

## 4. Discussion

Our results show that all three models considered here account for a significant portion of the cycle-to-cycle variance in the measured height change, particularly in the low-elevation, strong-melt subsamples of the data. However, the

MARv3.11.5 and GSFCv1.1 models both tend to overpredict total height changes by factors of up to around two, depending on the subsample of the data considered. These overestimates are most prominent in the Spring-Summer period in 2019, and in the strong-melt, high-elevation subsamples of the data, suggesting that melt processes play an important part in the overestimates. The updates to the melt model between GSFCv1.1 and GSFCv1.2 appear to improve these overestimates. Likewise, the rescaling experiments on the SMB and FAC showed that systematic rescaling of the magnitude of the SMB

processes in the model alone produced much smaller reductions in the residuals than systematic rescaling of FAC changes did, and for both MARv3.11.5 and GSFCv1.1, rescaling of FAC alone produced residual improvements approximately equal to those due to rescaling the total model or to rescaling the SMB and FAC separately. This points to melt of snow as the process most strongly driving the models' overestimates of height changes. In both MARv3.11.5 and the GSFC models, runoff is small over most of the ice sheet. This means that the SMB component of detrended height change is approximately equal to

positive contributions equal to the ice-equivalent snowfall, and negative contributions equal to the long-term average SMB rate that we subtracted to detrend the SMB. This component has relatively small temporal variability and cannot explain much of the variance in the heigh-change rate. In contrast to the SMB component, the FAC component has large temporal variations: When the surface of a snowpack begins to melt, the meltwater flows downward into the pore space in the snowpack, and until that pore space is full, the SMB change due to the melt is zero (because none of the melt runs off the ice sheet) and the total

model height change is equal to the FAC change. This means that any overestimate of melt over snow translates directly into

equal overestimates of surface-height change and FAC change. The MARv3.11.5 and GSFC models used different FDMs, but the melt for the GSFC models was based on a degree-day parametrization of the MARv3.5.2 melt. We expect GSFCv1.1 to share the MARv3 models' overestimates of height change, but in GSFCv1.2, the positive-degree-day scalings were limited for the high-elevation part of the ice sheet, which results in less total melt in this part of the ice sheet, and makes a notable

improvement in the model's performance during times when melt is large. We observe, however, that GSFCv1.2 does not fare better than the other models, and in fact has marginally larger residuals than MARv3.11.5 for the weak-melt subsamples of the data. Changes between GSFCv1.1 and GSFCv1.2 also include a different calculation of the initial surface density, which likely slightly increased the sensitivity of GSFCv1.2 total height change to melt events in the high-elevation interior of the ice sheet, and decreased it at low elevations. The improved model performance in regions where GSFCv1.2 was likely

more sensitive to melt events than GSFCv1.1 points again to better representation of melt in GSFCv1.2 as the major improvement between the GSFC models. The small reductions in residual spread that result from rescaling the SMB alone in the high-elevation part of the ice sheet for MARv3.11.5 and GSFCv1.1 (Fig. 7, A-B, Fig. 8, A-B) might provide weak evidence that the models overestimate SMB variability in this region, but the reductions in spread are much smaller than those associated with rescaling the FAC, suggesting that our analysis is not strongly sensitive to SMB scaling in this area.

The analysis in this study has focused on the variability of surface height at quarter-annual time scales. Any long-term differences between the modeled SMB/FD and the combined SMB, FD, and ice-flux divergence in the ice sheet will appear in our results as a non-zero mean residuals, caused by the regional mean of the differences, and as extra spread in the residuals, caused by spatial variability in the differences. Without additional information about the state of the ice sheet, we cannot distinguish the extent to which FD model errors (e.g. [*Helsen et al.*, 2008]), SMB-model errors, and errors in our

assumption that the ice sheet was in balance between 1980 and 1995 contribute the means and spreads in the residuals we measure. Despite this, the spread of the residuals to the best-fitting regressions (e.g. Fig. 7) bounds the spatial variability in any of these errors to ~decimeter scales or better in the ice-sheet interior, and to few-decimeter scales for elevations less than 2 km. The mean values of the corrected histograms also show clearly that the choice of models and scaling can make a substantial change in the interpretation of observed height differences. For example, tables S2-S4 show that although

subtracting the uncorrected MARv3.11.5 and GSFCv1.1 models leads to whole-ice-sheet estimates of dynamic change that are slightly positive for the spring-summer subsample of 2019 (~0.11 m for both models), subtracting the GSFCv1.2 model leads to a dynamic-change rate much closer to zero (0.028 m). This pattern holds for all subsamples of the data considered, where subtracting the GSFCv1.2 results in mean residuals closer to zero. Note that these means are not a good proxy for estimates of the total ice-sheet mass balance, because they are based on a weighted per-point average that is not guaranteed

(or especially likely) to produce a spatially uniform sampling of the surface area, and because the dynamic areas that we intentionally omitted from the study will have a strong influence on the total ice-sheet mass balance. Rather, they illustrate that the choice of FDM and SMB model is likely to have a large influence on recovered mass-balance estimates.

## 5. Conclusions

This study demonstrates one of the first applications of altimetry-difference data to the validation of surface-mass-balance and firn-densification models (and, to our knowledge, the first in Greenland). It demonstrates that the three models evaluated account for a large fraction of the observed height change in the low-elevation, high-melt areas of the ice sheet, but two of the three do not accurately account for the observed changes in higher elevation areas where melt is less common.

The results presented here are based on only two years' data, and we do not attempt to distinguish model errors from long-term ice-sheet mass imbalances. Consequently, we cannot reach firm conclusions about whether these models correctly represent long-term volume change rates for the ice sheet. In MARv3.11.5 and GSFCv1.1, The largest model-data differences appear to be associated with the representation of FAC changes in high-elevation parts of the ice sheet during melt events, which, for two reasons, should not necessarily imply errors in the long-term behaviour of the model. First, if the model densification is too rapid near the surface, the densification rates in the excessively dense firn should be slower at a later time, and the long-term mean densification rate may be largely correct. Second, until recently, high-elevation melt events were rare [*Trusel et al.*, 2018], so longer-term studies that use SMB/FD models to investigate decadal ice-sheet mass changes (e.g. [*Smith et al.*, 2020]) should see relatively small errors due to these events. Conversely, studies seeking to interpret ICESat-2 time series at seasonal time scales will need to account for errors in FD models to obtain accurate mass-change estimates. We note that for MARv3.11.5 and GSFCv1.1, residuals to the rescaled models tend to have means that are closer to zero than the unscaled models, suggesting that model errors may lead to more extreme (larger in absolute value) estimates of ice-sheet change due to ice dynamics.

Our results give little or no evidence for substantial errors in SMB rates in any of the models. Notably, rescaling SMB rates (Fig. 8) produces only marginal improvements in misfits beyond those from rescaling FAC. This is consistent with studies that have compared mass balance from ice-discharge and SMB models with gravimetric estimates of mass changes, which show consistent seasonal and interannual mass variations (e.g. [*Fettweis et al.*, 2020; *Sasgen et al.*, 2020]). We note that high-temporal-resolution gravimetric estimates of ice sheet mass change have been available for validation of SMB models for at least a decade, while seasonal altimetric measurements are relatively new, so it should not be surprising that the SMB models are better calibrated than the FD models. At the same time, the most significant deficiency that we infer in MARv3.11.5 and GSFCv1.1 is in the estimation of melt rates in the interior of the ice sheet, where meltwater is absorbed by the firn, and makes no contribution to runoff. If the same problem were to be present in models used to predict ice-sheet SMB in the future, when the climate is warmer and runoff is more prevalent at in the ice-sheet interior, we would expect them to predict excessively negative SMB rates.

Considered as a direct comparison of model accuracy, our results suggest that the most recent of the three models considered here, GSFCv1.2, has substantially smaller errors in representing surface changes in the high-elevation part of the ice sheet during melt events. Model improvements between GSFCv1.1 and GSFCv1.2 include changes in the initial density of new snow and a limitation in degree-day scaling factors in the melt model for high-elevation grid cells and based on the

improvement in unscaled model residuals at high elevations, we suggest that the latter made a substantial improvement in the model representation of surface-height change. Because GSFCv1.1 derives scaling factors based on melt and temperature data from an earlier MAR version (v3.5.2), and both models show similar behaviour during melt events in high-elevation regions, our observations confirm the suggestion that MARv3.11.5 likely overpredicts melt for the white-snow surfaces prevalent at high elevation in Greenland.

This study demonstrates a technique for directly evaluating surface-mass-balance model output using altimetry data. We propose that this has the potential to become a standard technique to allow modellers to test whether updates to model calculations or parameters improve model fidelity. Our study compared model versions that included a variety of different processes and parameters and were thus not designed to isolate the melt-driven height changes that we identified as needing improved representation in MARv3.11.5 and GSFCv1.1; a more targeted future study might include model experiments that change only a single parameter or process change at a time. We would also hope to see future studies include a larger variety of models, including, potentially, the popular RACMO-driven IMAU firn model [*Ligtenberg et al.*, 2018], and would hope to see the short-term densification information provided by altimetry studies fused with in-situ data such as firn strain measurements (e.g. [*MacFerrin et al.*, 2022]) and firn density profiles (e.g. [*Montgomery et al.*, 2018]) to produce holistic calibration of the short and long-term evolution of models.

**Data availability:**

ICESat-2 ATL11 data are available from the National Snow and Ice Data Center (NSIDC):
https://doi.org/10.5067/ATLAS/ATL11.004
MEASURES ice-velocity data are also available through the NSIDC:
https://doi.org/10.5067/OC7B04ZM9G6Q
The MAR model results are available by anonymous FTP:
ftp://ftp.climato.be/fettweis/MARv3.11/Greenland/ERA5_1950-2020-10km/daily_10km/
The GSFC model results are available through zenodo:
https://zenodo.org/record/7221954#.Y7N_I-zMLBs

**Author Contributions:**

Smith and Tedesco originally proposed the study, conceived and directed the research. Smith wrote most of the manuscript and developed the model-data comparison analysis. Medley and Fettweis provided model data, and provided interpretations of model-data discrepancies. Sutterley developed software for the study, and advised on the handling of SMB data. Alexander formatted and analysed model data; together with Porter, he provided insights into the behaviour of the models and suggested

numerical experiments. All authors participated in discussions of the manuscript and played a substantial role in manuscript writing and editing.

**Acknowledgments:**

The authors would like to thank the editor, Bert Wouters, and three anonymous referees for their efforts in helping to shape and refine our manuscript. We would also like to thank the ICESat-2 project, the ICESat-2 science team, and ATL11 developers Ben Jelley and Suzanne Dickinson.

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
