# Peer review of "Evaluating Greenland Surface-Mass-Balance and Firn-Densification Data Using ICESat-2 Altimetry"

_The Cryosphere, 2022_

## Referee Comment (RC3)

Smith et al. take up the challenge to assess the performance of combined SMB-FD models against laser-derived observations of elevation change over parts of the Greenland ice sheet where they assume ice-dynamical effects to be negligible. This is important, as it allows us to understand how to improve altimetry-based estimated of GrIS mass balance using firn and SMB models.

The paper is clearly written (in most places), and the scientific analyses are sound (in most places).

My only major critique of this paper is that the analysis of the dh correction, as well as the scaling experiments in section 3.2, are only presented in terms of histograms over the entire ice sheet, or over aggregated sections of the ice sheet.

Much more insight would be provided to the reader if time series were presented from a selection of locations across the ice sheet. What do time series of dh, dhm, dhc, dhFAC and dhSMB look like for an individual location in, for example, the lower western accumulation area, the southern interior, the northern interior, the northeast and the southeast? Rather than having to guess the physical reasons for improved agreement (reduced residuals), it would become clear at a process level from the time series.

As this is my only major concern, I encourage the authors to expand the paper to accommodate for it. It would strengthen further the discussion about the scaling experiments, because the authors will have figures with time series of dhFAC and dhSMB that immediately make obvious why scaling of dhFAC works and that of dhSMB won't.

Line by line comments:

L 29: heights vary -> elevation varies

L 38: perhaps good to clarify that you are referring to a climatologically mean surface mass balance here

L 51: This is a confusing statement. FD models are forced by meteorological parameters as well as by mass fluxes, both of which are computed by an SMB model.

L 68: Here you focus mostly on the densification part of an FD. However, the thermodynamical part of FD models is usually also evaluated against observations of deep temperature.

L 71: In Munneke et al. (2015), laser-observed dh/dt was tested against an FD model at selected locations in order to evaluate their model.

L 93: I assume that the separation of 3.3 km refers to the ground projection of the lasers, not of the lasers themselves

L 105: please reformulate: a strategy does not measure anything.

L 105: "At each of a set of reference points…" this sentence does not flow well

L 129: Why is it safe to assume that the errors derived from release-003 products are not too optimistic?

Table 1: Listing the internal model variables feels redundant since they are never referred to in the manuscript. This table can either be moved to the supplementary materials or removed entirely.

L 220: In 2008, Helsen et al. showed that systematic surface elevation change can be the delayed result of multi-decadal or even centennial variability in SMB. In the present setup of your study, this effect is not accounted for. Rather, like in other studies, changes are defined with respect to a reference period (in your case, 1980-1995) over which no change is assumed. However, in the interior over which you evaluate the SMB/FD models, any residual between observations and models could be caused by these very long-term effects originating from quite deep in the firn.

L 305 (figure 2): see major issue above. The 32 tiled maps are a very comprehensive way of presenting the data, but it lacks in detail, making it hard to judge the models against observations at key locations. My suggestion would be to add a figure with time series of dh, dhm and dhc for a few selected locations (e.g. west coast, southern interior, northern interior, NE coast, SE coast). In that way, it becomes much easier to appreciate the temporal simulation of elevation change by the models compared to the observations.

L 330 (figure 4): perhaps clarify here that the scaling factors X were defined such that dh - X dhm == 0

L 345: what does the scaling imply? Is surface density not sufficiently captured? Is there a structural overestimation of snowfall and/or melt? Is there a structural error in the ICESat observations?

L 425 (figure 7): the effect of only scaling dhFAC (light green line) is invisible in the graph.

L 425 (figure): why does rescaling the dhSMB make almost no difference, as opposed to rescaling dhFAC? Please elaborate on this.

L 456: why does it help to isolate errors in high-elevation melt when the agreement at lower elevations is good? Can we simply assume that an SMB model will perform well at higher elevations (snow albedo dominated) when it does so at lower elevations (ice albedo dominated)?

L 471: The elevation change in GFSCv1.2 is much less sensitive to melt events than the other two models. At the same time, its surface density has increased to 327-387 kg/m3, which is higher than the mean 315 kg/m3 reported by Fausto et al. in 2018. The surface elevation change associated with a melt event is approximately the amount of melt per unit area divided by the density of the melted snow: $\Delta h_{melt} \approx \Delta m_{melt}/\rho_{surface}$. Why do you think the higher surface density cannot explain the lower sensitivity of GFSCv1.2 to melt events?

L 514: GFSCv1.1 and GFSCv1.1 -> GFSCv1.1 and GFSCv1.2

---

## Author Comment (AC1)

**General remarks:**

We would like to thank all three referees for their kind words about our manuscript, and for the time that they took to read the manuscript, and to provide us with thoughtful suggestions for how to improve itt.  Reviewing manuscripts can be one of the most thankless jobs in our field, and It's important to acknowledge how important it is, and how useful it is for an author to get good and thorough reviews on a paper.  The recommendations from all three reviewers have helped us clarify some points that were obviously unclear in the initial manuscript, and have led us to clarify for ourselves (and for readers) a couple of scientific points that we had not fully thought through.

In our responses to the referees, we will be writing in blue, sans-serif font, justified to the left.  Quotes from the revised text will be in blue, in Times font, indented one stop.

**Referee 1:**

Although the ice-dynamic induced height changes (anomalies) can be neglected, how would the variations of local topography/roughness with (fast) ice flows affect the evaluation? This may have little impact for large-scale evaluation when the data are aggregated to a coarse resolution grid, but it would be good if the authors can comment/clarify on this point.

We have added a note to say that our velocity masking largely eliminates the large height-change signals that can be associated with advection of crevasses and rifts in the lower extremities of outlet glaciers.

Correction of firn compaction has been a critical step when using altimetry data to estimate the ice mass changes. RACMO has been more widely used in literature to correct for this effect. Although it may fall out of the scope of this study, it would be very helpful for the community if the authors can comment/discuss the RACMO firn estimates as well.

Adding an evaluation of RACMO to the study at this point would delay its publication more than we would like.  We have added this as a suggestion for future research directions in the last paragraph of our conclusions section.

Line 19. Specify the names of the three FD/SMB models evaluated in this study.

Done.

Line 22. Specify the names of the two models mentioned here.

Done.

Line 25. Specify the name of the third model here.

Done.

Line 186. Why did the authors use MARv3.5.2 for this step? How would the difference between MARv3.11.5 and MARv3.5.2 affect the evaluation? The reasons and potential biases should be clarified.

This is simply a matter of the history of when the GSFC models were calculated relative to the history of the MAR model releases.  The changes between MARv3.5.2 and MARv3.11.2 were not especially relevant to the surface melt rate in Greenland.  We now clarify this just after line 186

> "To derive a consistent melt-rate field, we used the MERRA-2 2-m temperatures as input to a degree-day model calibrated to MARv3.5.2 annual melt; the updates between MARv3.5.2 and the MARv3.11.2 model evaluated in this study did not have a major effect on temperature or  melt-rate estimates in Greenland, so we assume that the melt-rate calibration for the GSFC models is consistent with MARv3.11.2."

Section 2.3.1. This part (especially the first two paragraphs) is difficult to follow. Could the authors use some equations to explain the regression analysis done here?

We have split the 'regression analysis' section into two shorter sections, one describing the point weighting, the other describing the regression analysis.  We now provide a general equation for the regressions, and provide an example of how the regression works for the total model scaling, with a detailed description of the variables in the equation and of how to understand the results of the regression.  We hope that the additional description of our thinking and of how model parameters are used in the regression help clarify our thinking and make this part of the study easier to understand.

Section 3.2.3. This part is hard to follow too, with those scaling parameters and standard deviations. It would be helpful if the authors wrote some summary/topic sentences at the beginning of this section.

We have revised the text to put a summary of the most notable result described in each paragraph in the topic sentence, and have revised the body of each paragraph to omit some of the less relevant numerical values

Line 465. "..but the melt for GSFCv1.1 was based on a degree-day parametrization of the MARv3.11.5 melt..". Here is confusing. Did the authors use MARv3.11.5 or MARv3.5.2 to calibrate the degree-day model? MARv3.5.2 was mentioned in the methods part.

The earlier statement, that the GSFC models are based on MARv3.5.2 melt, is correct. We have amended the sentence at line 465 to read:

> The MARv3.11.5 and GSFC models used different FDMs, but the melt for the GSFC models was based on a degree-day parametrization of the MARv3.5.2 melt. We expect GSFCv1.1 to share the MARv3 model's overestimates of height change, but in GSFCv1.2, the positive-degree-day scalings were limited for the high-elevation part of the ice sheet, which results in less total melt in this part of the ice sheet, and makes a notable improvement in the model's performance during times when melt is large.

**Referee 2:**

ICESat-2 began measurements in October 2018. MacFerrin et al. recently published a firn compaction dataset, and I believe 2 of the sites have compaction measurements through 2019. It may be outside the scope of this study, but it may be interesting to compare ICESat-2 surface-height changes and modeled surface height changes at these two sites to examine the influence of firn compaction/atmospheric inputs to surface height changes and see how well the models capture these.

Thanks for the suggestion. We have tried comparing ice-surface-height records from weather stations with ICESat-2 and with FDM output, and found that there is noise and/or complexity in the single-point measurements of ice-surface height that made it difficult to compare them directly with other datasets. We think this might be a fruitful avenue to explore for other studies, but that it's beyond the scope of this one. We have added the suggestion to include strain-meter and firn density profiles to our final paragraph as a potential direction for future studies.

I would have liked to see the 3 models introduced earlier. It would be nice to list them in the abstract (e.g. Line 19) and in the introduction (e.g. Line 75).

Done.

It would be nice to clarify why you evaluate MARv3.11.5, but calibrate your degree-day parameterization in the GSFC model using MARv.3.5.2 (Sections 2.2.1 and 2.2.2).

Please see our response to referee 1 about line 186.

The regression analysis sections are quite detailed, and a bit difficult to understand (which may be my own problem). In Section 2.3.1, I did not quite get the point until you gave the example of scaling SMB by 0.5 (Line 254). It may be useful in this section to give a summary how the regressions are used for readers to then understand the more detailed methodology. I believe this would also be useful for Section 3.2.3.

We have moved the sentence where we describe how the expected regression results might relate to errors in the SMB or FD models to the start of the section, and have added some more discussion of different types of regression results. The start of section 3.2.2 (previously 3.2.1) now reads:

To help describe the relationship between modelled and measured height-change estimates, we calculate weighted regressions using components of the models' height changes as independent variables. Our goal in these regressions is to identify how the modelled height changes differ from the measurements over the ice sheet. These regressions estimate the scaling(s) for the model parameters that minimize the variance between the measured height differences and the sum of the scaled model parameters:

$$R_{model} = \sum W \left( dh - \left( dh_0 + \sum_{parameter\ j}^{\square} S_j dP_j \right) \right)^2 \qquad 1$$

Here $W$ are the point-density-based weights from 2.3.1, $dh$ are the measured height changes, $dP_j$ are differences in model parameters interpolated to the locations and times for the measurements that make up each height-change measurement, $S_j$ are the scaling values for each parameter, and $dh_0$ is the mean residual height change. The main statistic we use to evaluate the goodness of fit is the weighted standard deviation, calculated as:

$$\sigma = \left[ \frac{\sum r_i^2 W_i}{\sum W_i} \right]^{1/2} \qquad 2$$

Here $W_i$ are the inverse point densities, and $r_i$ are the regression residuals. As an example, in a regression between the total model height change and the observed height change (section 3.2.1) we solve for the coefficient, $A$, and the mean residual height change, $dh_0$, that minimize the quantity:

$$R_{model} = \sum W (dh - (dh_0 + A\ dh_m))^2 \qquad 3$$

Here $dh_m$ are the modelled height changes. Hypothetically, if one of the models were to systematically overestimate the surface mass balance by a factor of two, we would expect to see a regression for SMB result in a coefficient of 0.5 (meaning that scaling the SMB by 0.5 causes the model to fit the data), and residuals to that regression approximately equal to the data errors. Convserely, if the modelled height change was not strongly correlated with the measured height changes, we expect to see an arbitrar values for the regression coefficient, and to see a residual variance only slightly smaller than the data variance. Our analysis of the variance statistics is somewhat qualitative because we do not have a convincing way to determine the number of independent parameters in our regressions, but, as will be seen later in this paper, the distinction between variables for which regressions reduce the variance and those for which they do not is fairly clear.

Lines 282-284: Can you make this sentence clearer? You say "we can see that melt was considerably stronger in 2019 than it was in 2020". Can you specify where in the table we look to come to that conclusion? It is difficult to find in the table by keeping track of the variables.

Sorry for the confusion.  We have added a note to the end of this sentence:

(compare the f_melt statistics for sp-su 2019 with those from sp-su 2020).

Lines 466 and 517: Here you say that  melt for GSFCv1.1 was based on degree-day parameterization of the MARv3.11.5 melt. Earlier in the methods section you mentioned that it used MARv3.2.5. Could you clarify this?

Please see the response to referee 1's comment on line 465.  We also note at line 517 that the MAR version used in the calibration was 3.5.2

Line 500: What about using these models to predict ice sheet mass changes in the future using these SMB/FD models? It seems important that these models overpredict FAC changes associated with high-elevation melt events, which will likely be more frequent in the future.

This is a good point.  We have added the following to the end of the next paragraph (in which we discuss evidence for SMB errors in the models):

At the same time, the most significant deficiency that we infer in MARv3.11.5 and GSFCv1.1 was in the estimation of melt rates in the interior of the ice sheet, where meltwater is absorbed by the firn, and makes no contribution to runoff.  If the same problem were to be present in models used to predict ice-sheet SMB in the future, when the climate is warmer and runoff is more prevalent in the ice-sheet interior, we would expect them to predict excessively negative SMB rates.

Figures 4, 6, 7: Could you write out what each colored histogram represents in the figure caption? That would have benefited me while reading.

We have added the curve descriptions to the captions.

MacFerrin et al. citation:

MacFerrin, M. J., Stevens, C. M., Vandecrux, B., Waddington, E. D., and Abdalati, W.: The Greenland Firn Compaction Verification and Reconnaissance (FirnCover) dataset,

2013–2019, Earth Syst. Sci. Data, 14, 955–971, https://doi.org/10.5194/essd-14-955-2022, 2022.

We now cite this paper in our conclusions section, as part of our recommendation for further work.

**Referee 3:**

Smith et al. take up the challenge to assess the performance of combined SMB-FD models against laser-derived observations of elevation change over parts of the Greenland ice sheet where they assume ice-dynamical effects to be negligible. This is important, as it allows us to understand how to improve altimetry-based estimated of GrIS mass balance using firn and SMB models.

The paper is clearly written (in most places), and the scientific analyses are sound (in most places).

My only major critique of this paper is that the analysis of the dh correction, as well as the scaling experiments in section 3.2, are only presented in terms of histograms over the entire ice sheet, or over aggregated sections of the ice sheet.

Much more insight would be provided to the reader if time series were presented from a selection of locations across the ice sheet. What do time series of dh, dhm, dhc, dhFAC and dhSMB look like for an individual location in, for example, the lower western accumulation area, the southern interior, the northern interior, the northeast and the southeast? Rather than having to guess the physical reasons for improved agreement (reduced residuals), it would become clear at a process level from the time series.

As this is my only major concern, I encourage the authors to expand the paper to accommodate for it. It would strengthen further the discussion about the scaling experiments, because the authors will have figures with time series of dhFAC and dhSMB that immediately make obvious why scaling of dhFAC works and that of dhSMB won't.

We will include a figure that shows measured and modeled height changes for three locations that should be representative of the kinds of conditions we see around the ice sheet. We were initially optimistic that these plots would be a simple way of illustrating the processes at work in the SFM/SMB evolution, but the data did not cooperate especially well. Presenting the data as histograms and as thumbnail-sized maps of Greenland collapses a lot of small-scale variability in the models and in the data into figures that don't show the details of the process, but do illustrate the ice-sheet-scale statistics of the data and the models that we hope to explain in the paper.  When we select data and model results from small areas of the ice sheet, the local variability in the model  and the irregular sampling of the ATL11 data can produce plots that do not obviously tell the same story as do the aggregate statistics of the data over large areas.  Since the plots that we made take up

most of a page, it does not seem like a good use of journal space to include a lot of plots like this; instead, we discuss the general properties of the plots, and state that they are intended not as a representative sample of the data, but rather as an illustration of the small-scale structure of the measurements and models.

Line by line comments:

L 29: heights vary -> elevation varies

We understand the referee's discomfort with the use of "height" to describe the vertical location of the ice-sheet surface, when the broader community tends to use "elevation." This is a cultural quirk of the ICESat-2 community, which we hope all three referees can learn to forgive.

L 38: perhaps good to clarify that you are referring to a climatologically mean surface mass balance here

We added the words 'climatologically mean.'

L 51: This is a confusing statement. FD models are forced by meteorological parameters as well as by mass fluxes, both of which are computed by an SMB model.

We revised this sentence to say:

> FD models are driven by information about heat and moisture flux variability provided by SMB models,

L 68: Here you focus mostly on the densification part of an FD. However, the thermodynamical part of FD models is usually also evaluated against observations of deep temperature.

We now specify that it is the densification that is being tested in the studies we cite.

L 71: In Munneke et al. (2015), laser-observed dh/dt was tested against an FD model at selected locations in order to evaluate their model.

We added a reference to the Munneke study:

> We have identified one study [*Verjans et al.*, 2021] that has used altimetry differences to validate combined SMB and FD models in Antarctica, and a second [*Munneke et al.*, 2015] that used altimetry differences to evaluate trends in snow-surface heights predicted by models in Greenland.

L 93: I assume that the separation of 3.3 km refers to the ground projection of the lasers, not of the lasers themselves

The referee is correct. Instrument designs that included a 6.6-km wide spacecraft were deemed too expensive. We have edited the sentence to indicate that the measurements are separated by 3.3 km, not the lasers

L 105: please reformulate: a strategy does not measure anything.

(also for the next comment): We have deleted this paragraph, which was redundant to material in the next paragraph.

L 105: "At each of a set of reference points…" this sentence does not flow well

L 129: Why is it safe to assume that the errors derived from release-003 products are not too optimistic?

The study by Magruder et al., 2020, found that the actual geolocation errors on the product were better than 6.5 m, and that the nominal errors reported on the products (which were mostly meant to indicate that the errors had not been rigorously assessed at the time that the product was generated) were too large.  In any case, our statement in the text was too complicated, and we replaced it with this:

> However, because release-004 along-track products use nominal, pessimistic estimates of the geolocation errors (20 m in each direction), and studies that assessed the accuracy of release-003 products found that they in fact had smaller geolocation errors, generally  less than 6.5 m [*Magruder et al.*, 2020], we expect to see correlated errors ranging from a few cm in the interior to ~0.65 m in the most strongly (~10%) sloping areas near the coasts.

Table 1: Listing the internal model variables feels redundant since they are never referred to in the manuscript. This table can either be moved to the supplementary materials or removed entirely.

This is a reasonable suggestion. We will move table 1 to the supplementary material.

L 220: In 2008, Helsen et al. showed that systematic surface elevation change can be the delayed result of multi-decadal or even centennial variability in SMB. In the present setup of your study, this effect is not accounted for. Rather, like in other studies, changes are defined with respect to a reference period (in your case, 1980-1995) over which no change is assumed. However, in the interior over which you evaluate the SMB/FD models, any residual between observations and models could be caused by these very long-term effects originating from quite deep in the firn.

We agree with this statement, and have added the following text to the methods section:

> Although the spin-up of the FD model and our assumption of zero change during the reference period may result in errors in the detrended FD model results (e.g. [*Helsen et al.*, 2008]), we expect these errors to result primarily errors in the modelled height change that are steady over long (decadal) periods of time.  The quarter-annual height changes that are the main focus of this study may experience a temporally uniform shift (i.e. might all be too positive or too negative at a

particular location) as a result of these errors, but we do not expect the temporal variability of height changes to be significantly affected.

And have added the following text to the discussion section:

The analysis in this study has focused on the variability of surface height at quarter-annual time scales.  Any long-term differences between the modeled SMB/FD and the combined SMB, FD, and ice-flux divergence in the ice sheet will appear in our results as a non-zero mean residuals,  caused by the regional mean of the differences, and as extra spread in the residuals, caused by spatial variability in the differences.  Without additional information about the state of the ice sheet, we cannot distinguish the extent to which FD model errors (e.g. [*Helsen et al.*, 2008]), SMB-model errors, and errors in our assumption that the ice-sheet was in balance between 1980 and 1995 contribute the means and spreads in the residuals we measure.  Despite this, the spread of the residuals to the best-fitting regressions (e.g. Fig. 6) bounds the spatial variability in any of these errors to ~decimeter scales or better in the ice-sheet interior, and to few-decimeter scales for elevations less than 2 km

L 305 (figure 2): see major issue above. The 32 tiled maps are a very comprehensive way of presenting the data, but it lacks in detail, making it hard to judge the models against observations at key locations. My suggestion would be to add a figure with time series of dh, dhm and dhc for a few selected locations (e.g. west coast, southern interior, northern interior, NE coast, SE coast). In that way, it becomes much easier to appreciate the temporal simulation of elevation change by the models compared to the observations.

Please see our response to Referee 3's fourth paragraph.  We will add a figure demonstrating time series of model variables, height changes, and corrected height changes for three different locations on the ice sheet.  We feel that these plots are useful to provide context to the aggregated height-change maps and histograms, it is hard to capture a representative sample of how the models and data behave in plots of this type, and that the large-scale summary graphics are still the best way to illustrate what this study has learned about ice-sheet processes.  We hope that presenting both kinds of plots accomplishes what the referee was hoping for.

L 330 (figure 4): perhaps clarify here that the scaling factors X were defined such that dh - X dhm == 0

We have added notes to the *regression experiments* section to clarify how the regression parameters were calculated, and how the mean residuals and standard deviations were derived, which we hope satisfies this question.

L 345: what does the scaling imply? Is surface density not sufficiently captured? Is there a structural overestimation of snowfall and/or melt? Is there a structural error in the ICESat observations?

Line 345 is in the results section, so we do not provide much interpretation of the data. The discussion section includes answers to these questions. Our understanding (as explained in the results section) is that MARv3.11.5 and GSFCv1.1 systematically overestimate melt for high-elevation, white-snow conditions, a problem that is corrected in GSFCv1.2

L 425 (figure 7): the effect of only scaling dhFAC (light green line) is invisible in the graph.

We have added a note to explain this:

**Note that in each of these plots, the histogram for the separate rescaling of $dh_{FAC}$ and $dh_{SMB}$ is nearly identical to that for rescaling of $dh_{FAC}$ alone, so the histograms for $dh_{FAC}$ are not separately visible.**

L 425 (figure): why does rescaling the dhSMB make almost no difference, as opposed to rescaling dhFAC? Please elaborate on this.

This is an important point to capture, and we have added a comment explaining this to the discussion section:

In both MARv3.11.5 and the GSFC models, runoff is small over most of the ice sheet. This means that the SMB component of detrended height change is approximately equal to positive contributions equal to the ice-equivalent snowfall, and negative contributions equal to the long-term average SMB rate that we subtracted to detrend the SMB. This component has relatively small temporal variability, and cannot explain much of the variance in the heigh-change rate.

L 456: why does it help to isolate errors in high-elevation melt when the agreement at lower elevations is good? Can we simply assume that an SMB model will perform well at higher elevations (snow albedo dominated) when it does so at lower elevations (ice albedo dominated)?

We have deleted this sentence, whose purpose was somewhat unclear.

L 471: The elevation change in GFSCv1.2 is much less sensitive to melt events than the other two models. At the same time, its surface density has increased to 327-387 kg/m3, which is higher than the mean 315 kg/m3 reported by Fausto et al. in 2018. The surface elevation change associated with a melt event is approximately the amount of melt per unit area divided by the density of the melted snow: Why do you think the higher surface density cannot explain the lower sensitivity of GFSCv1.2 to melt events?

GSFCv1.2 is only less sensitive to melt events in the high-elevation part of the ice sheet. As it turns out, GSFCv1.2 is actually slightly less dense than v1.1 at high elevations, so the only available explanation is that the melt intensity is too high in v1.1. We have added a note to section 2.2.2 about the spatial distribution of density differences, and a note to the discussion section:

Changes between GSFCv1.1 and GSFCv1.2 also include a different calculation of the initial surface density, which likely increased the sensitivity of GSFCv1.2 surface height to melt events slightly in the high-elevation interior of the ice sheet, and decreased it at low elevations.  The improved model performance in regions where GSFCv1.2 was likely more sensitive to melt events than GSFCv1.1 points again to better representation of melt in GSFCv1.2 as the major improvement between the GSFC models.

L 514: GFSCv1.1 and GFSCv1.1 -> GFSCv1.1 and GFSCv1.2

Fixed.

---

## Referee Report (RR1)

I thank the authors for the work they have done to address the comments from the other reviewers and me on the previous version of the manuscript. The current work is a clear improvement and I recommend that the paper can be published after some minor edits aimed mostly at improving the clarity of the text.

Comments:

1. Section 2.3's wording was confusing to me. Do the authors calculate the block-median for the ICESAT-2 measurements or for the model results? With the sentence *'we assign the height differences into 2.5 km bins'*, do the authors means that they combined the ICESAT measurements into a map of 2.5 by 2.5 km grid cells? I recommend rewording this paragraph to improve its readability.

2. At line 235, the authors mention that they collected the accumulated melt for each model, but never use it in their analysis of their regression experiments. Why is this variable not used? Instead of making an educated guess about what resulted in the reported overestimation of dhFAC (densification rate, amount of melt, surface density), this variable could tell us what which process is responsible. For example, from figure 2 it seems that MAR overestimates melt in the low elevation regions, but GSFC underestimates melt. It would be nice if you could discuss this.

3. In figure 2, line 235 and line 325 the model total height is mentioned, but also the surface height. Are these variables different from each other? If so, how are they different? If not, I recommend using only one or the other term.

4. The authors use 'height differences' and 'height changes' interchangeably. This was especially confusing while reading the methods section. Please use 'changes' to indicate measured or modelled surface elevation changes and reserve the word differences for when you are comparing ICESat-2 with the model data.

5. The results for the high-elevation subset of your data suggests that there is an overestimation of snowfall in the interior part of the ice sheet (fig 7A-C and 8A-B) in GSFC 1.1 and MAR (and possibly also GSFC 1.2) because the regression leads to a lower dhm and dhSMB. This may be worth pointing out in the text.

---

## Author Response (AR2)

Response to comments on TC-2022-44, second round.

We thank Bert Wouters and the two anonymous referees for their further attention to this manuscript. We have taken almost all of the suggestions into account, and are glad to have the opportunity to clarify some of the remaining confusing aspects of our study. I have interposed our responses to Dr. Wouters and to the referees in blue: Our dialog is shown in times new roman font, and any quotes from the revised manuscript are shown in *italics*.

--Ben

**Editor's comments:**

**Comments to the author**:
Dear Ben and co-author,

The referees have commented on your revised manuscript and are satisfied with the changes that were made. Your manuscript is now almost ready for publication, but I would like you to make the following, final changes:

• Be consistent in use of "ice sheet" when not an adverb (e.g. line 283: "For the ice-sheet as a whole" -> "For the ice sheet as a whole"). Use height changes for temporal height changes in the observations/models, and height differences when comparing observations and models (R3)

We corrected three instances where we had "ice-sheet" in place of "ice sheet." Thanks for noticing

• Split the sentence starting on line 26 in two (second sentence starting with "The third model…"

Done.

• Provide references for the statements on lines 37-39.

Please see our note on R1's comment. It appears to be standard practice in the literature to state that vertical stretching of ice associated with horizontal flux divergence is part of the local mass balance of the ice sheet without providing a reference. We added a parenthetical note to explain what we meant.

• Fig. S3: use "GSFCv1.1 and v1.2" instead of "GSFC v1.1 and v1.2.1"

Done

• Lines 243-246 was unclear to R1, who seems to interpret the changes in point density to be dependent on elevation. It may help to explain that there are more measurements in the North than in the South due to the converge of the ground tracks, i.e. latitude dependent (unless I'm misinterpreting the sentence…)

It sounded to us as if R1 was reading this as a paragraph about snow density. We made sure to refer to "measurement density" to help avoid further confusion.

• Add units on line 292, and explain what z_melt is (this may also address the comment of reviewer 3 concerning not using accumulated melt from the models).

Fixed

• Figure 2: explain in caption why some of the heights are (not) connected by solid lines (I assume the unconnected dots are from the first two cycles?)

You are correct about the crossovers, plus there are data gaps later on. We added a note to the caption explaining this.

• Check if reference on line 370 to figure 4 is correct. Include references to figures 4C and 4K, and to figure 4E/M on line 372, as suggested by R1.

Fixed

• Line 373: mention explicitly which season you are referring to.

Fixed.

• Clarify what the difference is between "total height" and "surface height", or stick to one of the two.

Fixed.

• R3 notices that the accumulated melt is computed but not used in the analysis. If z_melt is indeed equal to the accumulated melt (see previous comment), this problem is solved. Otherwise, remove reference to accumulated melt on line 236.

We fixed table S1 and made sure to define z_melt as 'accumulated melt' in the text.

• The start of section 2.3 was confusing to R3, I suggest to change this to:

We reduce the full set of 57 million height-difference measurements from ICESAT-2 to a more compact sample with a more even spatial distribution by calculating a block-median set of height differences for each cycle-to-cycle (~91-day) epoch. For each epoc, we group the height differences into 2.5 km grid cells for each ICESat-2 pair track. For each such cell in each epoch, we identify the measurement (or measurements) that matches (or bracket) the median height difference. For each model, we then record the model parameters (i.e. surface height, model total height, model SMB anomaly, model235 FAC, and model accumulated melt) corresponding to the median value. For each median difference measurement, we sample each of our model fields at the location and time of the height measurements, and calculate their differences. This gives a set of model-field-difference values that are precisely collocated with the measured differences.

We made (approximately) this modification. Thanks for offering a clear rewrite of the paragraph.

Furthermore, the referees have made suggestions which I believe would further strengthen the paper, but I leave the choice on implanting these to you.

Please see our responses to the referees below.

- L32: May be more succinct to say "mass changes of ice sheets".

Fixed

- Section 2.3.2: You can take this or leave it, but it may be helpful to make the two dh's clearer by calling one dh_meas and the other dh_mod. I know currently you distinguish them by dh and dh_m, but it may be more helpful for readers to keep track with more descriptive subscripts? If you choose to implement this change, make sure it is changed throughout (e.g. Figure 5, 7, 8).

This is a good idea, but would require substantial modifications to the text, the equations, and the figures. With regret, we are not following this suggestion

- The results for the high-elevation subset of your data suggests that there is an overestimation of snowfall in the interior part of the ice sheet (fig 7A-C and 8A-B) in GSFC 1.1 and MAR (and possibly also GSFC 1.2) because the regression leads to a lower dhm and dhSMB. This may be worth pointing out in the text.

Please see my response to referee 3's point 5. The residual reductions for the dhSMB rescaling are very small, and don't provide good evidence one way or the other.

- Instead of making an educated guess about what resulted in the reported overestimation of dhFAC (densification rate, amount of melt, surface density), this variable could tell us what which process is responsible. For example, from figure 2 it seems that MAR overestimates melt in the low elevation regions, but GSFC underestimates melt. It would be nice if you could discuss this.

This question refers to the z_melt variable. We didn't include an explicit regression WRT z_melt because errors in z_melt do not have a consistent relation to surface-height changes. We had tried something like this in developing the study, but its interpretation was not straightforward.

Congratulations on a very interesting paper!

Kind regards
Bert Wouters

Comments from referee 1:

I would like to thank the authors for revising the manuscript to improve the clarity and allow readers to better understand the methodology used. I recommend this

manuscript be published subject to minor technical corrections. Below, I list just a few suggestions that I think would help to further improve clarity of the paper. The authors can decide whether they choose to implement the changes or not – none of them will impact the scientific findings of the paper. Congratulations on this impressive study!

Thanks for the kind description of the study, and for the recommendations.  We realize that this manuscript is a bit dense in places, and are glad that Dr. Wouters was able to find referees who were game to go through it carefully.

Throughout the text, you use both "ice-sheet" and "ice sheet". This should be made consistent. I would recommend using "ice sheet".

We had a look through the paper, and made sure that we were using "ice-sheet" as an adjective, and "ice sheet" as a noun.  I don't think we ever used it as an adverb (which would have no hyphen).

L27: by using "and" here in "and the third model…" makes it seem like GSFCv1.2 exacerbates the overprediction. I would suggest changing this to something like "however, the updated high-elevation melt prediction in GSFCv1.2 avoids this overprediction." Or something along those lines.

We split this sentence into two:

*This overprediction seems to be associated with the melt sensitivity of the models in the high-elevation part of the ice sheet. The third model, GSFCv1.2, which has an updated high-elevation melt parameterization, avoids this overprediction.*

L32: May be more succinct to say "mass changes of ice sheets".

Fixed.

L37-39: Could you provide some references here? The SMB expected change is obvious, but may be nice to provide some citations for variation in the local stress balance.

To us, this statement seemed somewhat axiomatic (note that the statement is about flux divergence, not about stress) so we examined a few other papers that treat the relationship between horizontal velocity and ice-sheet mass balance, and didn't see that any of them offering a citation for a similar statement.  We added a parenthetical example to help avoid confusion:
 *On an ice sheet in steady state, whose volume and mass are constant in time, snow accumulation and ice ablation at the surface are balanced by ice-flux divergence in the ice-snow column (i.e., by thinning or thickening of the ice column related to horizontal stretching of the ice),*

L243-246: This new sentence is a little unclear to me. The distribution of density differences is not spatially uniform, but to me, it seems better to say that the density differences are spatially coherent (or elevationally-coherent, or something like this) – density increases are concentrated in the low-elevation ice sheet periphery, while density decreases are found in the high-elevation interior.

I think the referee misread this paragraph: The density in question here is the number of measurements per unit area, not the physical density of the snow.  Reading the paragraph over, it seems that what's written is

reasonably clear, but to further help avoid confusion, we made sure to always refer to "measurement density" or "measurement-density values."

Fig S3: Should the caption use "1.2" rather than "1.2.1" to remain consistent with the main text?

Fixed.

Section 2.3.2: You can take this or leave it, but it may be helpful to make the two dh's clearer by calling one dh_meas and the other dh_mod. I know currently you distinguish them by dh and dh_m, but it may be more helpful for readers to keep track with more descriptive subscripts? If you choose to implement this change, make sure it is changed throughout (e.g. Figure 5, 7, 8).

This is a good idea, but would require substantial modifications to the text, the equations, and the figures. With regret, we are not following this suggestion

Line 292: Add units denoting the subsets. E.g. (h < 2000 m).

Fixed.

Line 292: Maybe make it clear where the z_melt comes from (I assume it is the model accumulated melt). You have mentioned model parameters before, and that one is model accumulated melt, but then all of a sudden the z_melt variable appears. You could just say "into strong-melt subsets using the model accumulated melt parameter, z_melt, …."

We now specify that z_melt is accumulated melt, and have fixed this in table 1.

Figure 2: Very minor comment – but why are some of the heights from different RPTs in panels D-O connected by solid lines, while others are not?

Comment added to the caption:

*Time series for each RPT are joined by a solid line when derived from continuous repeat-track measurements. Broken lines or lone points indicate crossover measurements or missing values in the repeat-track measurements.*

Line 370: I think you are now referencing Figure 4 here since you added a new figure above? Also, I would reference Fig. 4C, 4K after you say "resulting in positive corrected values".

Fixed

Line 372: Add reference to Figure 4 (maybe panels E,M?) to illustrate the overestimates of thickening during colder seasons.

Fixed

Line 373: "During that same season"… I assume you are talking about Q2 of 2019? I

would explicitly say that here, because the previous sentence ends with you discussing the colder seasons, and I don't see readily apparent large discrepancies in the bare ice zones between models during this time.

Fixed

**Comments from Referee 3**

I thank the authors for the work they have done to address the comments from the other reviewers and me on the previous version of the manuscript. The current work is a clear improvement and I recommend that the paper can be published after some minor edits aimed mostly at improving the clarity of the text.

Thanks for the recommendation, and for the further attention to the manuscript. These changes will be quite helpful to the readers.

Comments:
1. Section 2.3's wording was confusing to me. Do the authors calculate the blockmedian for the ICESAT-2 measurements or for the model results? With the sentence 'we assign the height differences into 2.5 km bins', do the authors means that they combined the ICESAT measurements into a map of 2.5 by 2.5 km grid cells? I recommend rewording this paragraph to improve its readability.

Sorry for the confusion! We rewrote this paragraph as follows (with thanks to B.W. for his recommended changes):
    For each model, we reduce the full set of 57 million height-difference measurements from ICESAT-2 to a more compact sample with a more even spatial distribution by calculating a block-median set of height differences for each cycle-to-cycle (~91-day) epoch. For each epoch in each model, we assign the height differences into 2.5 km cells for each ICESat-2 pair track. For each such cell in each epoch, we identify the measurement (or measurements) that match (or bracket) the median height difference. For each median difference measurement, we sample each of our model fields (i.e. model total height, model SMB anomaly, model FAC, and model accumulated melt) at the time and location of the height measurements, and calculate their differences. This gives a set of model-field-difference values that are precisely collocated with the measured differences.

2. At line 235, the authors mention that they collected the accumulated melt for each model, but never use it in their analysis of their regression experiments. Why is this variable not used? Instead of making an educated guess about what resulted in the reported overestimation of dhFAC (densification rate, amount of melt, surface density), this variable could tell us what which process is responsible. For example, from figure 2 it seems that MAR overestimates melt in the low elevation regions, but GSFC underestimates melt. It would be nice if you could discuss this.

In developing the paper, we did regressions against z_melt, but did not find a concise way to interpret the results. We added an excuse for not describing these regressions around line 300 in the revised manuscript:
    We perform regressions for the total model change ($dh_m$), the height change due to SMB anomalies ($dh_{SMB}$), and the height change due to firn-air content ($dh_{FAC}$). We use the modelled total melt to segregate the data into strong-melt ($z_{melt} > 0.2 |dh_m|$) and weak-melt ($z_{melt} < 0.2 |dh_m|$) subsets, but do not perform explicit regressions between surface-mass-balance change and melt because melt does not have a consistent linear relationship with surface-height change. The height change associated with melt depends on the density of the snow or ice being melted, and on whether the meltwater runs off the ice sheet or is refrozen,

*which makes the results of a regression between $z_{melt}$ and dh more difficult to interpret than those for the other variables.*

3. In figure 2, line 235 and line 325 the model total height is mentioned, but also the surface height. Are these variables different from each other? If so, how are they different? If not, I recommend using only one or the other term.

Our inclusion of "surface height" in the list of parameters was a mistake. We looked through the rest of the paper, and where we were referring to something predicted to the model, used the term "total height."

4. The authors use 'height differences' and 'height changes' interchangeably. This was especially confusing while reading the methods section. Please use 'changes' to indicate measured or modelled surface elevation changes and reserve the word differences for when you are comparing ICESat-2 with the model data.

We acknowledge the confusion that using two different terms can cause, and made some wording changes to be more consistent. We now use "height differences" and "difference measurements" to refer to what we measured with the satellite, and use "height changes" to refer to what happened on the ice sheet, and what the models predicted. We refer to what is obtained when a modelled height change is subtracted from a measured height difference as a "residual." We hope that this makes our results easier to follow.

5. The results for the high-elevation subset of your data suggests that there is an overestimation of snowfall in the interior part of the ice sheet (fig 7A-C and 8A-B) in GSFC 1.1 and MAR (and possibly also GSFC 1.2) because the regression leads to a lower dhm and dhSMB. This may be worth pointing out in the text.

This is true, but we feel that the evidence for an overestimate of snowfall is not very strong. The changes dh_m in these conditions results in very small changes in the residuals compared to what we see when we vary the FAC component.
We have added a note at 538-541:

*The small reductions in residual spread that result from rescaling the SMB alone in the high-elevation part of the ice sheet for MARv3.11.5 and GSFCv1.1 (Fig. 7, A-B, Fig. 8, A-B) might provide weak evidence that the models overestimate SMB variability in this region, but the reductions in spread are much smaller than those associated with rescaling the FAC, suggesting that our analysis is not strongly sensitive to SMB scaling in this area.*

---

## Author Response (AR3)

Dear Bert,

We have made the changes you recommended, and caught one additional mistake:

Line 36:  removed sentence fragment

Lines 55 and 57:  Fixed references

Line 63: maxed -> masked.

Changed table 2 to table 1 (table 1 is now in the SOM)

Thanks again for all your help!

Best
Ben and co-authors